# FlashAttention on a Napkin: A Diagrammatic Approach to Deep Learning IO-Awareness

**Vincent Abbott**  *Vincent.Abbott.24@ucl.ac.uk*
*Department of Computer Science, University College London*

**Gioele Zardini**  *gzardini@mit.edu*
*Laboratory for Information and Decision Systems, Massachusetts Institute of Technology*

**Reviewed on OpenReview:** *https://openreview.net/forum?id=pF2ukh7HxA*

## Abstract

Optimizing deep learning algorithms currently requires slow, manual derivation, potentially leaving much performance untapped. Methods like FlashAttention have achieved a $\times 6$ performance improvement over native PyTorch by avoiding unnecessary data transfers, but required three iterations over three years to be developed. Automated compiled methods have consistently lagged behind. This paper extends *Neural Circuit Diagrams* for deep learning models to consider resource usage and the distribution of tasks across a GPU hierarchy. We show how diagrams can use simple relabellings to derive high-level streaming and tiling optimization strategies along with performance models. We show how this high-level performance model allows the effects of quantization and multi-level GPU hierarchies to be readily considered. We develop a methodology for representing intermediate-level pseudocode with diagrams, allowing hardware-aware algorithms to be derived step-by-step. Finally, we show how our methodology can be used to better understand existing techniques like FlashAttention. This work uses a theoretical framework to link assumptions about GPU behaviour to claims about performance. We aim to lay the groundwork for a scientific approach to GPU optimization where experiments can address clear hypotheses rather than post-hoc rationalizations.

## 1 Introduction

### 1.1 Background

To execute an operation, graphical processing units (GPUs) must move data from high-level DRAM to low-level compute cores. GPUs are as limited as much by GB/s of memory bandwidth as TFLOPs of available compute. However, AI models have passed the *memory wall*—algorithms are increasingly limited by bandwidth/transfer costs (Ootomo & Yokota, 2023; Ivanov et al., 2021; Gholami et al., 2024), as compute capability has improved far more quickly $\times 3/2$y than DRAM bandwidth $\times 1.6/2$y (Gholami et al., 2024). Furthermore, DRAM already accounts for 46% of total system power (Ghose et al., 2018). As memory becomes increasingly inefficient relative to compute, the importance of considering transfer costs—*IO-awareness* (Dao et al., 2022; Aggarwal & Vitter, 1988)—will become even more critical.

FlashAttention (Dao et al., 2022; Dao, 2023; Shah et al., 2024) is an IO-aware approach to attention that overcomes the memory wall. Attention (Vaswani et al., 2017) is central to generative models, including large language models (LLMs) (Mistral AI team, 2024; Llama team, 2024) and image generation algorithms (Ho et al., 2020; Esser et al., 2024; Rombach et al., 2022; Podell et al., 2024). FlashAttention *fuses* the steps of attention. It computes all sequential steps on low-level memory, avoiding unnecessary intermediate data transfers. It achieves a $\times 6$ increase in throughput compared to standard PyTorch, arguably making large contemporary models possible.

However, the conditions under which fusion is possible are not generally exploited. Simple cases like element-wise functions can be compiled into matrix multiplications (**?**Paszke et al., 2019; Sabne, 2020), but the bespoke techniques of FlashAttention required manual derivation and three iterations over three years to take full advantage of Hopper hardware (NVIDIA, 2022) features. *Triton* (Tillet et al., 2019) offers some compilation for hardware features but has lagged behind new FlashAttention algorithms (Dao, 2023; Shah et al., 2024). The current best technique for generating IO-aware algorithms that exploit hardware features remains slow, manual derivation.

Innovating new optimized algorithms is essential to efficient model deployment. In addition to FlashAttention, methods like grouped query attention (Ainslie et al., 2023), KV-caching (Shazeer, 2019), and quantization (Frantar et al., 2023; Gholami et al., 2022) all reduce transfer costs while having minimal impact on the function we implement or the quality of model outputs. Much like fusion, the success of these approaches relies on understanding the compositional structure of algorithms so that similar but less costly algorithms can be executed. A systematic approach to innovating optimized algorithms will require a mechanism for understanding the compositional structure of algorithms along with a performance model which compares varying means of executing the same operation.

The hardware characteristics of GPUs have a significant impact on performance which varies depending on the target algorithm. When choosing between A100s, H100 SXM5s, or H100 PCIes (NVIDIA, 2022, p.39), we must consider the varying compute, bandwidth, intermediate hierarchies, architecture features, and low-level memory, for which we pay in environmental and economic resources. The application of these features is often non-obvious, FlashAttention-2 (Dao, 2023) was released while the hardware for FlashAttention-3 already existed (Shah et al., 2024), which achieved $\sim 75\%$ improvement in forward speed. Understanding the impact of GPU features is a necessary component of innovating optimized approaches and making full use of deployed resources.

## 1.2 Contributions

This paper contributes a representational scheme for deep learning algorithms based on *Neural Circuit Diagrams* that shows the distribution of tasks across a GPU hierarchy and associated resource usages (Section 2). This scheme incorporates theorems about the compositional properties of fused algorithms (Appendix A.1), allowing for the rapid derivation of high-level sketches of GPU optimized matrix multiplication and attention, along with performance models (Section 3). This performance model can consider quantization and multi-level hierarchies (Section 4), bolstered by theorems (Appendix A.2). With a model corresponding to a Hopper-like architecture, we use a step-by-step process to derive hardware-aware algorithms (Section 5). These derivations can consider coalesced memory access, tensor-core operations, and overlapping operations, and reveal the degrees of freedom in the algorithm's configuration. Our methodology can be used to analyse the bottlenecks of existing methods like FlashAttention (Section 6).

The main value of this work is providing a potential framework for relating assumptions about GPU behavior to claims about performance. This allows future empirical work to address and improve specific assumptions, thereby iteratively improving the model. This contrasts with the prevailing "tinkering" approach within deep learning that develops post-hoc intuition based on past experiments. Post-hoc intuition is trained on the "test set" of prior successful approaches, weakening generalizability and scientific value. Our diagrammatic scheme allows the rapid application of the framework to algorithms used in practice, distinguishing it from descriptive theory and toy model approaches. Diagrams leverage human visual comprehension and enable complex theorems and underlying implications to be encapsulated with simple notation.

## 2 Diagramming Deep Learning Algorithms

### 2.1 Diagramming Functions and Data Types

Diagrams have alternating columns of data types and functions. Data type columns are shown in Figure 1. Arrays such as $\mathbb{R}^{a \times b \times c}$ are represented by a wire for each axis labeled with the size. Data types may be tuples of arrays, such as $\mathbb{R}^{a \times b \times c} \times \mathbb{R}^{d \times e \times c}$, and are represented by placing a dashed line between constituent arrays.

Figure 1: We represent arrays, of forms such as $\mathbb{R}^{a \times b \times c}$, by labeling stacked wires in a column with $a$, $b$, and $c$. To represent data types that consist of lists of arrays, such as $\mathbb{R}^{a \times b \times c} \times \mathbb{R}^{d \times e}$, we place a dashed line between them.

The diagram represents the array type $\mathbb{R}^{a \times b \times c}$ by stacking the labeled wires.

Stacking with a dashed line assembles the data type $\mathbb{R}^{a \times b \times c} \times \mathbb{R}^{d \times e}$.

Functions between data types are represented by labeled boxes or pictograms with their input/output shapes to the left/right. Sequential execution (*composition*) of functions is shown by horizontal placement (Figure 2, creating a diagram with alternating columns representing data types and functions. Parallel execution (*concatenation*) of functions stacks them with a dashed line in between (Figure 3). A concatenated function takes concatenated inputs and provides concatenated outputs. The change in the input/output is reflected by the diagram.

Figure 2: Functions are represented by labeled boxes or pictograms, which aid intuition. These representations can be horizontally composed, which represents sequential execution and yields a diagram with alternating data type and function columns. We represent composition by $F; G = G \circ F$.

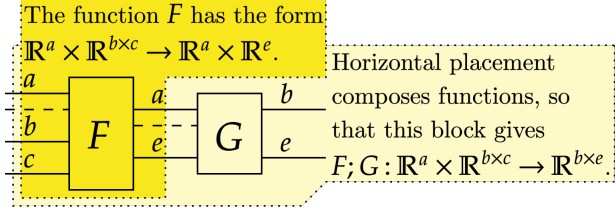

The function $F$ has the form $\mathbb{R}^a \times \mathbb{R}^{b \times c} \to \mathbb{R}^a \times \mathbb{R}^e$.

Horizontal placement composes functions, so that this block gives $F; G : \mathbb{R}^a \times \mathbb{R}^{b \times c} \to \mathbb{R}^{b \times e}$.

Figure 3: Functions can also be stacked with a separating dashed line, which concatenates their inputs and outputs. Concatenating tuples $\otimes$ is considered to be associative. For concatenated functions $F \otimes G$, if $F(x) = x'$ and $G(y) = y'$ then $(F \otimes G)(x \otimes y) = F(x) \otimes G(y)$.

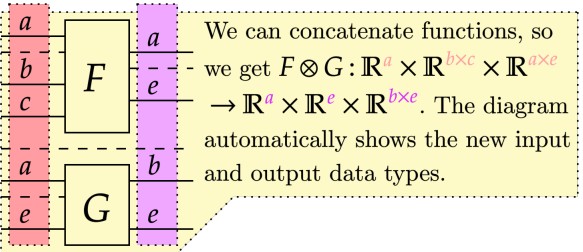

We can concatenate functions, so we get $F \otimes G : \mathbb{R}^a \times \mathbb{R}^{b \times c} \times \mathbb{R}^{a \times e} \to \mathbb{R}^a \times \mathbb{R}^e \times \mathbb{R}^{b \times e}$. The diagram automatically shows the new input and output data types.

We represent identity functions that leave inputs unchanged by extending the data type. This reflects that composition with identities leaves a function unchanged. Functions are stateless and are defined by how they map inputs to outputs. Therefore, we concatenate with identities to represent a function acting on some data but leaving the rest unchanged and available. With these tools, we can build compound diagrams such as Figure 4.

Figure 4: A compound diagram can be disassembled into columns representing alternating functions and data types. Stacked functions and data types can be further decomposed to find the core units concatenated to construct them. Identities are represented by continuing the representation of data types.

We have alternating columns of data types and functions. These columns are concatenated along dashed lines. Continued wires in function columns are identities.

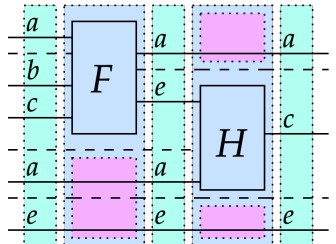

Functions can be mapped over an additional axis, represented by weaving the axis over the function's outputs and a choice of the inputs (Figure 5). This diagrammatic implementation naturally updates the input and output sizes for the mapped function. When an input segment is not weaved, its data is copied to evaluate each index along the outputs of the new axis. The axis can be weaved into any location of the target segments.

SoftMax
*on a vector* $\mathbb{R}^x$

SoftMax
*on rows of* $\mathbb{R}^{q \times x}$

Contraction
*for vectors* $\mathbb{R}^b$

Matrix Multiplication
*of* $\mathbb{R}^{a \times b} \times \mathbb{R}^{b \times c}$

Figure 5: A function can be weaved, which adds an axis to the outputs and some of the inputs. The function is mapped over this axis. When we weave the "item" $\mathbb{R}$ array represented by a thick dotted wire, we can remove it. Here, we provide a weaving for SoftMax, represented by a triangle, to have it act over each row of an array, and of linear contraction (*dot/inner product*), which provides matrix multiplication.

Weaving a function allows for complex mappings to be represented and avoids the ambiguity of typical expressions. We can weave primitives defined on items, such as multiplication, addition, and copying. We use weaving to represent the splitting and joining of shared axes, which overcomes the typical difficulties of expressing how arrays are partitioned and concatenated. We show this in Figure 6.

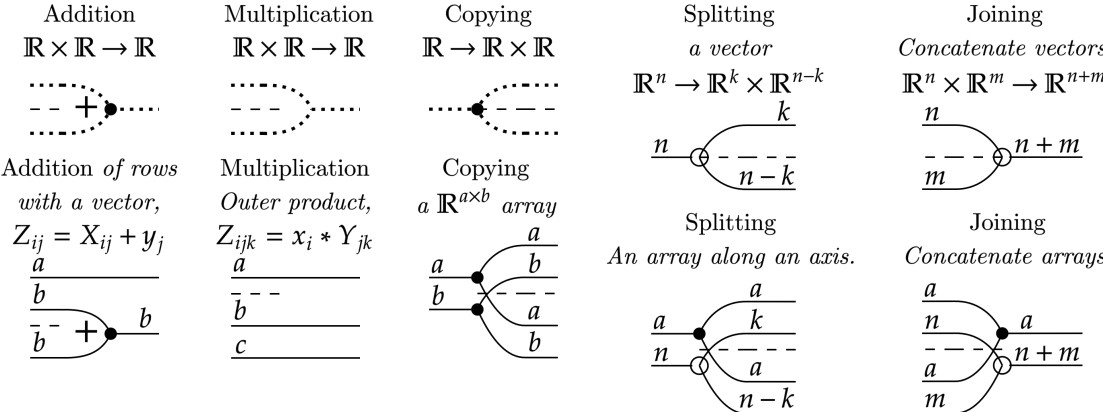

Figure 6: Weaving primitive functions let us express the addition of vectors to each row of an array, multiplication over a specific axis, and the copying of arrays. We can use weaving to split an array along slices of an axis or show the axes over which arrays are joined.

## 2.2 Representing Deep Learning Algorithms

We have so far expressed *functions* — maps between inputs and outputs — diagrammatically. Deep learning models employ *algorithms*, which implement a function but have resource usages and inputs/outputs located at different levels. We embed algorithms in a hierarchy. A hierarchy consists of levels connected with pipes (as in Figure 7), which allow for memory sharing with a family of cores located at the level below. The available algorithms are restricted to those provided at each level of the hierarchy.

H100 SXM5 GPUs consists of 132 streaming multiprocessors (SM), each with compute capabilities. The GPU has global GMEM memory, two L2 caches, and shared memory/register files (SMEM/RMEM) per SM. Logically, this is organized as a *grid* (device-wide) of *threadblocks* (limited to an SM) consisting of *warps* (which pipeline instructions) each with 32 *threads*. Threadblocks have independent allocations of SMEM, and threads have independent allocations of RMEM. We abstract these details away for our hierarchy, focusing on the logical organization and memory sharing capabilities.

Figure 7: We can diagram hierarchies using a graph showing the available levels and their connections. These hierarchies model real GPU, and provide levels corresponding to logical abstractions.

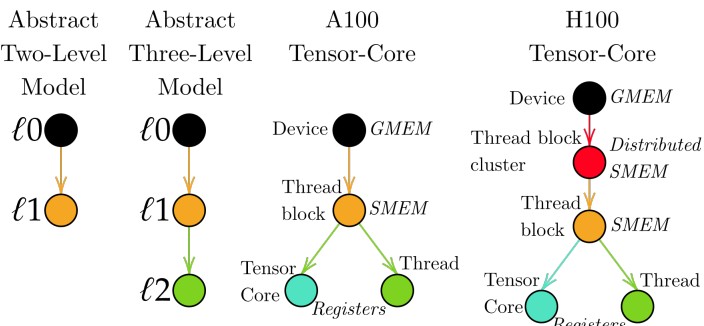

We use colors to represent levels, and color arrays to diagram where they are located. In this section, we use a two-level model where higher level $\ell0$ arrays are colored black and lower level $\ell1$ arrays are colored orange. This lets us diagram algorithms as in Figure 8.

Figure 8: Here, we diagram an algorithm which takes a SoftMax over a $\overline{q} \times \overline{x}$ array and contracts it over a $\overline{x} \times \overline{d}$ array. We perform transfers to move data to lower levels for computation. This diagram shows sequential execution, concatenation, and weaving of algorithms diagrammatically.

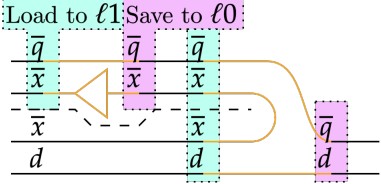

We are interested in algorithms' total transfer cost $H_\ell$ and maximum memory usage per core $M_\ell$ (memory usage). These resource usages are defined per level and measured in number of values, and can be determined from diagrams as in Figure 9. Total transfer costs are equal to the total size of data loaded to and saved from a level, equal to the sum of the size of arrays changing colors. Memory usage is lower bounded by the maximum size of data at a level for any column. We aim to minimize the total transfers while keeping memory usage below a hardware limit per level, $M_\ell^{\max}$.

Figure 9: The SoftMax-Contraction algorithm from Figure 8 can have its transfer cost derived from the total size of data changing colors and its memory usage lower bound determined by the maximum data present at the lower level at any point. We assume that $\overline{x} > d$.

| $M_{\ell1}$ | | | $\overline{q}\overline{x} + \overline{x}d$ | | *Cumulative* |
|---|---|---|---|---|---|
| $H_{\ell1}$ | $\overline{q}\overline{x}$ | $\overline{q}\overline{x}$ | $\overline{q}\overline{x} + \overline{x}d$ | $\overline{q}\overline{x}$ | $4\overline{q}\overline{x} + \overline{x}d$ |

As diagrams show all data types, operations, and their arrangement, we can adapt our performance model to consider all aspects of an algorithm's performance. Using diagrams, we can approximate compute by taking the compute required to execute an algorithm multiplied by the size of axes it is weaved over. A $k$-size contraction requires $2k$ FLOPs; therefore, $m \times k$ by $k \times n$ matrix multiplication requires $2mkn$ FLOPs. In Section 5.7, this is used to find the clock cycles required per column to overlap computation.

### 2.3 Group Partitioning

The first optimization strategy we introduce is group partitioning (*tiling*) a mapped algorithm (Figure 10). If an algorithm is weaved over an axis, then the axis can be split, the mapped function applied, and the axis rejoined to yield the same result. Each sub-algorithm acts on a batch of the data in a separate core with reduced low-level memory usage.

Figure 10: A weaved algorithm is functionally equivalent to sub-algorithms acting on partitions of the weaved axis. We use $\equiv$ to indicate functional equivalence, meaning algorithms map the same inputs to outputs but may have distinct resource consumption profiles, and therefore are not strictly equivalent. These partitions can be of any size, which we write as $g_a$. We can recursively expand the expression on the right until $a' \leq g_a$. The unweaved segment of the data must be loaded by each sub-algorithm.

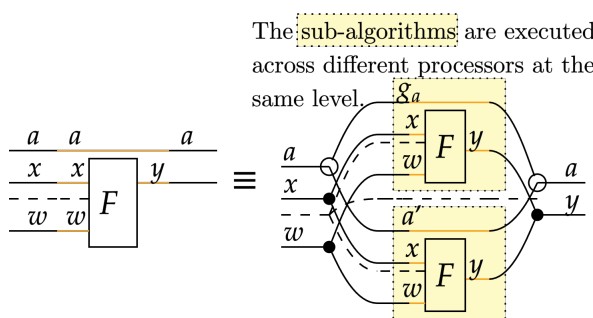

We can diagram this strategy by labeling the weaved axis $a$ with the target group size $g_a$ while at the lower level as in Figure 11. The low-level memory usage $M_\ell$ for the diagram is then calculated using this group size, not the full size of the axis. Each sub-algorithm needs to load and save its batch input and output data. The per-group transfer cost $H_{\ell,g}$ is calculated using the $g_a$ group size for the partitioned axis which is multiplied by $N_{\ell,g} = a/g_a$ batches to attain $H_\ell = N_{\ell,g} H_{\ell,g}$.

Non-grouped inputs are sent to all active cores at the lower level, meaning their transfer costs are multiplied by $N_{\ell,g}$ without reduced per-group transfer costs. Smaller group sizes $g_a$ decrease memory usage but increase $N_{\ell,g}$, increasing $H_\ell$ if there is an unweaved input. To reduce total transfer costs, we must find the maximum $g_a$ value that does not exceed maximum memory usage $M_\ell^{\max}$.

Figure 11: Group partitioning can be represented by relabeling an algorithm with the partition batch size (group size) at the lower level. The group size is used for memory usage and per-group transfer cost calculations for the lower level. Per-group transfer costs are then multiplied by the number of groups $N_{\ell,g} = a/g_a$ to give the overall transfer costs.

| | | | Cumulative |
|---|---|---|---|
| $M_{\ell 1}$ | $g_a x + w$ | $g_a y$ | |
| $H_{\ell 1,g}$ | $g_a x + w$ | $g_a y$ | $g_a(x+y) + w$ |

$$M_{\ell 1} \geqslant g_a x + w, g_a y$$
$$H_{\ell 1,g} = g_a(x+y) + w$$
$$N_{\ell 1,g} = a/g_a$$
$$\therefore H_{\ell 1} = ax + ay + aw/g_a$$

If multiple weaves are relabeled, the data is batched over each as in Figure 12. The total number of sub-algorithms is the product of the number of batches for each axis, $N_{\ell,g} = ab/g_a g_b$. The relabeled sub-algorithm represents the memory usage and per-group transfer costs of each sub-algorithm, using the group sizes $g_a$ and $g_b$ for resource usage calculations. We then multiply the transfer costs by $N_{\ell,g}$. We can use this to determine the optimal group sizes for an algorithm grouped over multiple axes, such as matrix multiplication (see Section 3.1), and to determine whether it is worth grouping a small axis or transferring its full size.

Figure 12: A function with multiple weaves can have a relabeling applied to each of its weaves. The relabeled sub-algorithm provides the memory usage and per-group transfer costs using the group sizes.

| $M_{\ell 1}$ | $g_a x + w g_b$ | $g_a y g_b$ | Cumulative |
|---|---|---|---|
| $H_{\ell 1, g}$ | $g_a x + w g_b$ | $g_a y g_b$ | $g_a x + g_b w + g_a g_b y$ |

$$M_{\ell 1} \geqslant g_a x + g_b w, g_a g_b y$$
$$H_{\ell 1, g} = N_{\ell 1, g}(g_a x + g_b w + g_a g_b y)$$
$$N_{\ell 1, g} = ab / g_a g_b$$
$$\therefore H_{\ell 1} = abx\left(g_b^{-1} + g_a^{-1}\right) + aby$$

## 2.4 Stream Partition

Stream partitions (*recomputation*) exploit recursively decomposable polymorphic functions to feed data in batches while maintaining intermediate outputs on-chip, reducing low-level memory usage. Functions can be streamed if they are polymorphic for a specific axis (defined for that axis being of any size) and have an accumulator that can incorporate incoming data to recompute the maintained output, as shown in Figure 13. This allows for a recursive expansion (see Figure 14) that maintains minimum data on-chip at any point.

Figure 13: The condition for streaming requires that a function $F$ be polymorphic along the axis $a$ and can be decomposed in the manner above, requiring the existence of another polymorphic function $B$ called the accumulator.

Figure 14: If the condition in Figure 13 is met, the function can be recursively decomposed until $a' \leq s_a$. This allows the function to be evaluated from batches of the input data.

If an axis originates from a transfer and is fed to a recursively decomposable polymorphic function, then it can be relabeled with the streaming batch size (stream size) $s_b$ as in Figure 15. This creates a representation of the sub-algorithm $B$ which is repeatedly applied to process the data. We need to add the output size $y$ in parentheses at the input to consider its contribution to memory usage. The memory usage of the algorithm is then determined using the stream size $s_b$ instead of the full axis size $b$. As we eventually stream the entire axis, we use the full axis size $b$ to evaluate transfer costs. Typically, we strictly benefit from limiting the stream size to 1 as this reduces memory usage while imposing no increase in transfer costs.

Figure 15: We can relabel a streamable axis with the batch size $s_b$ as it is transferred. We are required to add the $y$ output array shape at the input to the streamed algorithm. This lets the relabeled diagram derive the memory usage at the lower level using the stream size $s_b$. As all data along the axis is transferred to the chip, the full axis size $b$ must be used for transfer costs.

| $M_{\ell 1}$ | $a s_b + w + y$ | $y$ | Cumulative |
|---|---|---|---|
| $H_{\ell 1}$ | $ab + w$ | $y$ | $ab + w + y$ |

$$M_{\ell 1} \geqslant a s_b + w + y$$
$$H_{\ell 1} = ab + w + y$$

Per the fusion theorems of Appendix A.1, streamable axes are resistant to modifications. The streamable axis may be a single axis of an array, and composing or weaving a streamable algorithm while maintaining this axis yields a streamable algorithm. This allows the stream labeling to be maintained for resource usage evaluation as in Figure 16, and allows the streamability of complex functions like attention to be derived from a streamable kernel as in Figure 21. In Figure 17, we apply group partitioning to a mapped streamable algorithm. We use $g_q$ for per-group transfer evaluations, and both $g_q$ and $s_b$ to evaluate memory usage.

Figure 16: For a modified streamed algorithm, we can continue to use the stream batch size $s_b$ for memory usage evaluations. As the function generates $q \times r$ distinct $y$ values, it needs to maintain each on memory, resulting in $y \times q \times r$ maintained memory before and after the repeated $E$ algorithm.

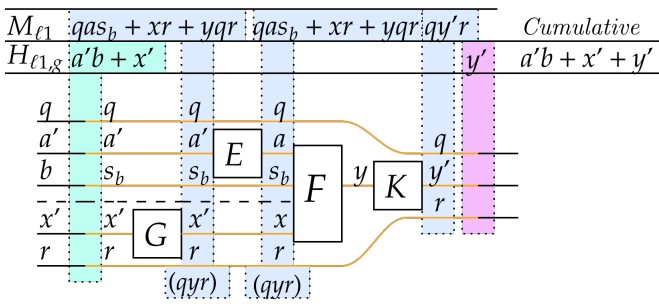

Figure 17: We can apply multiple relabelings to an algorithm. This lets us find the per-group memory usage and transfer cost. As the function is mapped within each group, it needs to maintain $g_q$ copies of the maintained $y$ data, increasing its memory usage.

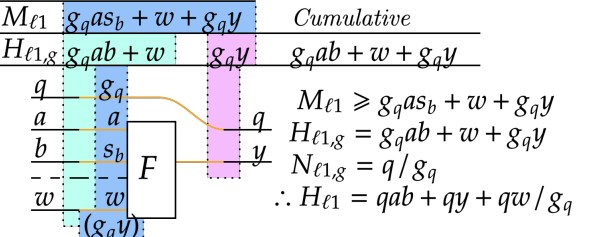

## 3 Examples

### 3.1 Matrix Multiplication

As contraction (dot product) is streamable (see Appendix A.3.1), we can use it as a kernel for deriving the streamability of matrix multiplication, its weaved form. This provides a diagram that supplies a performance model. We then optimize for the batch sizes to minimize total transfers given some maximum lower-level memory usage $M$.

$$N_g = \frac{ac}{g_a g_c} \qquad H = N_g H_g$$

$$H_g = g_a b + b g_c + g_a g_c \qquad = \frac{ac}{g_a g_c}(g_a b + b g_c + g_a g_c)$$

$$M \geqslant g_a g_c + g_a s_b + s_b g_c \qquad = abc(g_c^{-1} + g_a^{-1}) + ac$$

$$\sqrt{M} \geqslant g_c = g_a \qquad \geqslant 2abc\, M^{-0.5} + ac$$

Figure 18: The dot product is a streamable function. Therefore, matrix multiplication, which is the weaved form of it, is also streamable and can be group partitioned.

This simple derivation reveals an immense deal about matrix multiplication and its associated performance model. Firstly, we have derived that $g_a$ should equal $g_c$, corresponding to a square tiling shown in Figure 18. We see that bandwidth increases with $b$ at a rate dependent on the size of memory. A standard approach would effectively assume that $M \to \infty$. This would cap $g_a$ at $a$ and $g_c$ at $c$, not the memory limit, yielding $H = ab + bc + ac$ (Gholami et al., 2024; Ootomo & Yokota, 2023). Considering the tiling size and memory constraints, we get $H \geq 2abcM^{-0.5} + ac$. In Appendix B.1, we examine the arithmetic intensity.

Furthermore, we can use matrix multiplication as an instructive preview of the next section which focuses on multi-level performance models. We can compute each $g_a \times s_b \times g_c$ matrix multiplication at a lower, green level. As square tiling of $g_a = g_c$ is optimal for sizes $a$ and $c$, square tiling of $h_a = h_c$ is optimal for sizes $g_a$ and $g_c$. This generates a recursive tiling pattern which allows the two-level model generated by simple diagrams like Figure 18 to be extended to complex GPU hierarchies.

Figure 19: Figure 18 represents a tiling process where we split the $a \times b$ matrix into tiles of size $g_a \times s_b$ and the $b \times c$ matrix into tiles of size $s_b \times g_c$. Each lower-level processor is assigned responsibility for a $g_a \times g_c$ of the output, for which it accumulates outputs by iterating down $s_b$. The effect of $g_a$ and $g_c$ on total transfers is reflected in the corresponding equations of Figure 18, implying a square tiling is optimal.

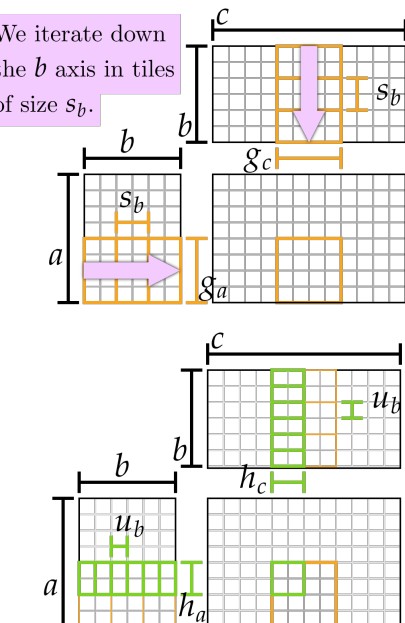

Figure 20: If there is a green level below the orange, then the $g_a \times g_c$ responsibility tiles are further split into blocks of size $h_a \times h_c$, each assigned to a lower-level processor. Each $s_b$ stream loaded to the orange level can be split into $u_b$ streams for the lower level. Responsibility assignment from the black to orange levels is similar to responsibility assignment from the orange to green levels, allowing us to recursively extend the two-level model. This algorithm is diagrammed in Figure 24.

## 3.2 Attention

We derive the streamability of attention from the fusion theorems. We begin with the fact that SoftMax-Contraction is streamable (Appendix A.3.2). Then, we can compose with a contraction over the queries as an $E$ algorithm from Figure 16. This generates a streamable algorithm, which we weave with the $\overline{q}$ and $d$ axes. This generates Figure 21. Correctness is ensured as the diagram gives the typical expression for attention, $O = \text{SoftMax}\left(Q \cdot K^T\right) \cdot V$, with axes clearly indicated. We can then label $\overline{q}$ to distribute the queries across processors, yielding the FlashAttention algorithm. Figure 21, then, can be seen as deriving and providing a performance model for FlashAttention.

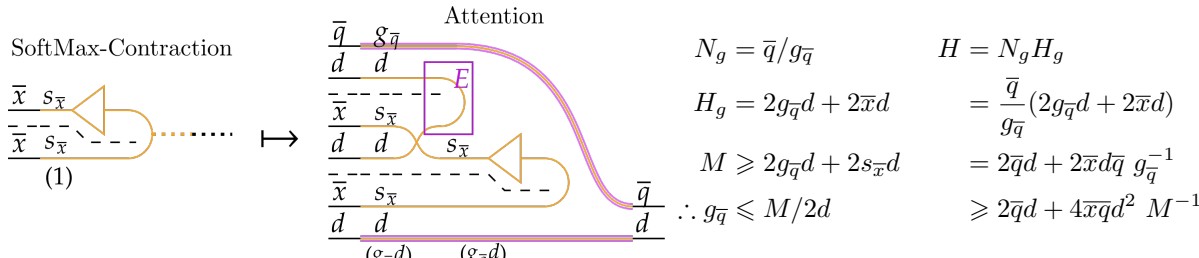

Figure 21: SoftMax followed by a contraction is streamable. We precompose with a contraction $E$ weaved by $s_x$ and provide weavings by $\overline{q}$ at the top and $d$ at the bottom to construct attention.

We can apply a similar technique to find the transfer cost of grouped query attention (Ainslie et al., 2023) (Figure 23) and multi-head attention (Vaswani et al., 2017) (Figure 22). These use additional weaves, but their evaluation remains straightforward. This shows how diagrams can be used to both derive optimizations and experiment with modifications to the algorithm, motivating further innovation.

Figure 22: Multi-head attention conducts multiple attention algorithms in parallel. It is represented by an additional $h$ axis weaved over every operation. Notice how scaling $h$ leads to a linear change in costs, while $d$ leads to a quadratic change.

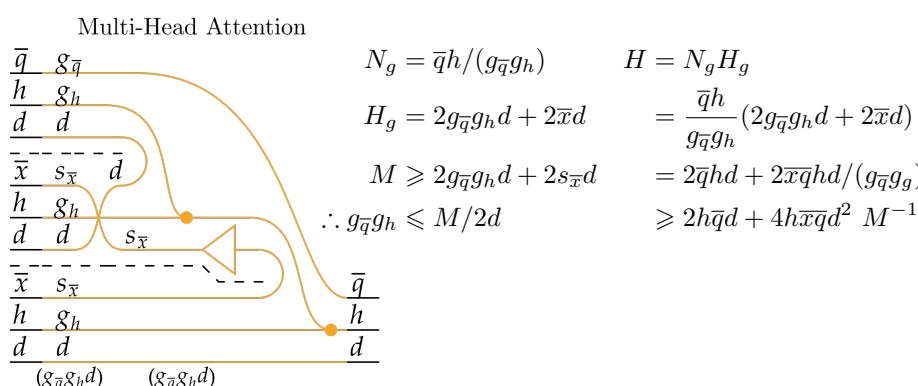

Multi-Head Attention

$$N_g = \overline{q}h/(g_{\overline{q}}g_h) \qquad H = N_g H_g$$
$$H_g = 2g_{\overline{q}}g_h d + 2\overline{x}d \qquad = \frac{\overline{q}h}{g_{\overline{q}}g_h}(2g_{\overline{q}}g_h d + 2\overline{x}d)$$
$$M \geqslant 2g_{\overline{q}}g_h d + 2s_{\overline{x}}d \qquad = 2\overline{q}hd + 2\overline{x}\overline{q}hd/(g_{\overline{q}}g_g)$$
$$\therefore g_{\overline{q}}g_h \leqslant M/2d \qquad \geqslant 2h\overline{q}d + 4h\overline{x}\overline{q}d^2 \; M^{-1}$$

Figure 23: Grouped query attention has an additional weave accompanying the queries. This reduces the required parameters and compute to generate the keys and values. However, we observe that the required transfers are similar to full multi-head attention.

Grouped Query Attention

$$N_g = g\overline{q}/(g_{\overline{q}}g_g) \qquad H = N_g H_g$$
$$H_g = 2g_{\overline{q}}g_g d + 2\overline{x}d \qquad = \frac{g\overline{q}}{g_{\overline{q}}g_g}(2g_{\overline{q}}g_g d + 2\overline{x}d)$$
$$M \geqslant 2g_{\overline{q}}g_g d + 2s_{\overline{x}}d \qquad = 2g\overline{q}d + 2g\overline{x}\overline{q}d/(g_{\overline{q}}g_g)$$
$$\therefore g_{\overline{q}}g_g \leqslant M/2d \qquad \geqslant 2g\overline{q}d + 4g\overline{x}\overline{q}d^2 \; M^{-1}$$

## 4 Analysis of Performance Models

Once a two-level model optimization of an algorithm is found, we can extend it to consider a multi-level hierarchy. Each lower level has tiles which fit into the level above (Figure 24), meaning the optimal strategy and performance model extend in a generalizable manner. We can create universal performance model for transfer costs which considers the impact of the GPU hierarchy and the transfer rate and memory caches at different levels. This allows us to make informed choices between different GPU architectures given their energy and capital costs, levels of quantization we employ, and the configuration of GPU hierarchies.

### 4.1 Optimal Transfers $H^*(\vec{a}, M)$

Applying the two-layer model to diagrams provides optimal transfer costs $H^*(\vec{a}, M)$ given some configuration of axis sizes $\vec{a}$ and lower-level memory $M$. So far, these expressions have a standard form given by the sum of power functions which solves for Equation 4 in Appendix A.2:

$$H^*(\vec{a}, M) = \sum_t \alpha_t(\vec{a}) \; M^{-\beta_t} \tag{1}$$

The index $t$ iterates over terms, the coefficient $\alpha_t(\vec{a})$ is dependent on the axis sizes, and $\beta_t \geqslant 0$ is an exponent greater than zero as transfers necessarily decrease with increased memory size. The exponents $\beta_t$ indicate the sensitivity of performance to memory size and indicate how data is distributed. For attention, the $M^{-1}$ factor indicates the data is broadcast to all groups, while for matrix multiplication the $M^{-0.5}$ factor indicates square tile distribution.

## 4.2 Multi-Level Performance Models

An algorithm with multiple levels requires $H^*(\vec{a}, M_\ell)$ transfers for each. Even though data cannot be directly transferred from the highest to lowest levels, lower levels can utilize the data loaded and saved by intermediate levels. The execution of $H^*(\vec{a}, M_{\ell 1})$ intermediate level transfers makes data available for the lower level $\ell 2$ and accounts for saving it back. Each intermediate-level tile fits a larger number of low-level tiles (see Figure 20), among which it distributes its data. This fitting process has a negligible error with large $M$. We assign a weighted transfer cost $\dot{H}_\ell^{-1}$ to each level. For the highest level, we assume $M_{\ell 0} \to \infty$ and $\dot{H}_{\ell 0}^{-1} = 0$, as data is already present. This means that the total weighted transfer cost of an algorithm can be expressed by:

$$H^* = \sum_\ell \dot{H}_\ell^{-1} \; H^*(\vec{a}, M_\ell) = \left( \sum_t \alpha(\vec{a}) \; \sum_\ell \dot{H}_\ell^{-1} \; M_\ell^{-\beta_t} \right) \tag{2}$$

Therefore, the relative performance of GPUs is determined not just by the raw transfer rates but also by the memory size of available levels and the specific algorithm being implemented. For attention, the key factor per level is $\sim \dot{H}^{-1}/M$, while matrix multiplication has $\dot{H}^{-1}/M^{0.5}$. This implies that attention is more sensitive to the size of lower-level memory and that less lower-level memory (smaller tile sizes) is required to avoid bandwidth constraints.

## 4.3 Quantization

Equation 2 and the two-level model consider transfers and storage limits in terms of number of values. However, GPUs are restricted by the number of bytes we can transfer and store. If we have $q$ bytes per value, then the maximum number of values $M_\ell = M_\ell^{\text{Bytes}}/q$ and the transfer weight is $\dot{H}_\ell^{-1} = (\dot{H}_\ell^{\text{Bytes}}/q)^{-1}$. Substituting these expressions into the total transfer cost, we get:

$$H^{*\text{Bytes}} = \sum_\ell (\dot{H}_\ell^{\text{Bytes}}/q)^{-1} \; H^* \left( \vec{a}, M_\ell^{\text{Bytes}}/q \right)$$

$$= \left( \sum_t \alpha(\vec{a}) \; \sum_\ell (\dot{H}_\ell^{\text{Bytes}})^{-1} \; (M_\ell^{\text{Bytes}})^{-\beta} \; q^{1+\beta} \right) \tag{3}$$

As $1 + \beta \geqslant 1$, total transfers are superlinear to the degree of quantization. Halving the quantization from $FP32$ to $FP16$ can accelerate attention by up to $\times 4$, and improves large matrix multiplication by a factor of $2^{1.5} \approx 2.83$. This indicates that a generous use of quantization and specialized kernels is critical to high-throughput implementations of models.

## 4.4 Intermediate Caching

We can choose to store output data at lower levels, and save it up in chunks. This changes the level immediately above the lower level to a caching level, which we indicate by adding asterisks to its output data as in Figure 24. The size of this column is not memory restricted by the intermediate level which is only used to temporarily store data as it is sent up. However, the lower levels must remain active to store data, and this imposes the restriction that $N_{g,\ell 2}/N_{g,\ell 1} \leqslant N_{\ell 2}^{\max}$ which is a hardware limit. With an output restricted algorithm, this results in $H^*(\vec{a}, N_{\ell 2}^{\max} M_{\ell 2})$ transfers being required for the intermediate level $\ell 1$, using the total lower level memory $N_{\ell 2}^{\max} M_{\ell 2}$ instead of its own hardware maximum memory $M_{\ell 1}^{\max}$. This is elaborated in Appendix A.2.1.

## 4.5 Cross-Transfer Levels

Our model can encompass levels that perform inter-core communication instead of providing shared memory by using modified weighted transfer weights. These cross-transfer levels encompass H100 thread block clusters (Luo et al., 2024), multi-GPU systems, and intra-warp communication. We set up $h$ as a higher level, $x$ as a cross-transfer level, and $c$ as a child level composed of linked processors. Instead of sending $H^*(\vec{a}, M_c)$ data directly to children, we send $H^*(\vec{a}, M_c N_c^{\max})$ data to any of the interconnected children and cross-transfer

Figure 24: With multiple levels in a hierarchy, we can maintain output or streamed data at the lowest levels and use the intermediate levels as a cache. The cache size does not contribute to the $M_{\ell 1}^{\max}$ restriction. Instead, we require that $N_{\ell 2,g}/N_{\ell 1,g} \leqslant N_{\ell 2}^{\max}$. If an algorithm is output restricted, we can implement this by the cache size being less than $M_\ell = M_{\ell 1} N_{\ell 2}^{\max}$.

| | | | |
|---|---|---|---|
| $M_{\ell 1}$ | $g_a s_b + s_b g_c$ | $g_a g_b$ | |
| $M_{\ell 1}$ | $h_a s_b + h_b g_c + h_a h_c$ | $h_a h_c$ | *Cumulative* |
| $H_{\ell 1,g}$ | $g_a b + b g_c$ | $g_a g_b$ | $g_a b + b g_c + g_a g_c$ |
| $H_{\ell 2,g}$ | $h_a b + b h_c$ | $h_a h_c$ | $h_a b + b h_c + h_a h_c$ |

$$M_{\ell 1}^{\max} \geqslant g_a s_b + s_b g_c$$
$$M_{\ell 2}^{\max} \geqslant h_a h_c$$
$$N_{\ell 2}^{\max} M_{\ell 2}^{\max} \geqslant g_a g_c$$
$$\Leftrightarrow \ N_{\ell 2,g}/N_{\ell 1,g} \leqslant N_{\ell 2}^{\max}$$

the remaining data as in Figure 25. This results in a performance model with modified transfer weights and levels, adding a level $x$ between $h$ and $c$ with transfer weight $\dot{H}_{h \to c}^{-1} - \dot{H}_{x \to c}^{-1}$ and memory $M_c N_c^{\max}$, and replacing the transfer weight of level $c$ with $\dot{H}_{x \to c}^{-1}$. We outline this derivation in the Appendix A.2.2.

Figure 25: To send data to children, we directly transfer $H^*(\vec{a}, M_c N_c^{\max})$ distributed across the child processors, treating the cross-transfer level as having memory of size $M_c N_c^{\max}$. We then perform the remaining $H^*(\vec{a}, M_c) - H^*(\vec{a}, M_c N_c^{\max})$ transfers as fast cross-transfers.

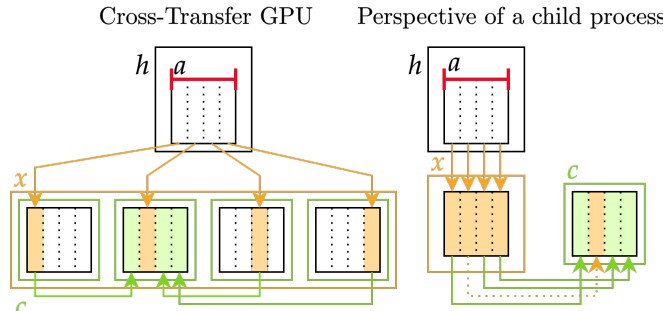

# 5  Pseudocode and Hardware Optimizations

So far, we have focused on abstract models that represent the generic assignment of tasks across a GPU hierarchy and investigated the accompanying performance model. This systematic methodology provides the same level of detail as the theorems from FlashAttention (Dao et al., 2022).

The aim of the diagrammatic approach is to quickly develop optimized kernels. The success of FlashAttention-2 (Dao, 2023) and FlashAttention-3 (Shah et al., 2024) relied on considering specialized hardware features; tensor core operations and asynchronous compute (in the case of Hopper). Standard methods required years of exploration to incorporate these aspects. With diagrams, we can systematically identify the contexts in which hardware features can be applied and quickly derive kernels.

In this section, we develop the tools to go from generic, abstract analysis to developing an algorithm for a specific hardware architecture. In this section, we provide a systematic procedure to go from an abstract two-level model to an algorithm configured for a specific GPU architecture. We will work with a toy hierarchy which imitates Hopper, with levels for the different modes in which memory can be stored; SMEM, registers, or tensor cores. Available subalgorithms will be dictated by the functionalities of the respective levels. The aim of this section is to not focus on Hopper specifically, as many aspects of implementation will be missed. Instead, it is to show how diagrams can be used to systematically derive a hardware-aware algorithm. This methodology can be extended to Ampere, Blackwell, and non-NVIDIA architectures.

**Coalesced Memory Transfer**

Between the device and SMEM memory, GPUs move data in chunks of $128B$ of consecutive memory. This is remarkably straightforward to represent with diagrams. Arrays represent how data is distributed across each stride, so the lowermost axis of an array represents consecutively stored memory. If we enforce that the lowermost axis is divisible by $128B/q$ when assembled in the device and transferred between GMEM and SMEM, then we can assure coalesced memory access. This may require that each thread block stream loads

data for multiple lower-level streams. If using SMEM as a cache, there is usually plentiful memory available for larger streams.

A floating divisor in the superscript of an axis/batch size is used to indicate a value it is divisible by (see Figure 26). This is done at the point where the restriction is imposed and along the immediately weaved axis. Multiple divisors impose the least common multiple.

**Tensor Core Operations**

Tensor cores provide very fast matrix multiplication, and their consideration was the major contribution of FlashAttention-2 (Dao, 2023). Modern GPUs have far more FLOPs available for tensor cores than general-purpose register operations (NVIDIA, 2020; 2022). However, tensor cores require memory to be fragmented in an incoherent manner across warps (groups of 32 threads) or warpgroups (groups of 128 threads, Hopper only) which makes data unsuitable for general-purpose operations. We can still perform element-wise operations, as these do not require information about the location of data. As a result, tensor cores act like a distinct level of the memory hierarchy, having access to a different set of algorithms than general-purpose thread data. Coherent SMEM data can be fragmented for tensor cores, providing a pipe between the threadblock and tensor core levels.

Tensor cores (`wmma` or `wgmma`) can only manage data at certain sizes and quantizations (NVIDIA, 2024) (`wmma`, `wgmma`). Quantization can be represented by a floating tag. Matrix multiplications of larger sizes can be implemented by adding multiple smaller matrix multiplications, making divisibility the critical factor. We can enforce this restriction by placing superscripts for tensor core axes, as shown in Figure 26.

Figure 26: Multi-level matrix multiplication uses the SMEM level to cache data for lower-level tensor core operations. We enforce the divisibility restrictions for coalesced SMEM transfers and tensor cores using superscripts.

## 5.1 From Diagrams to Pseudocode

We can expand streamed algorithms into looped pseudocode forms where all variables are explicitly shown as in Figure 27. The columns of pseudocode diagrams provide the size of variables required in memory and the transfers/operations we need to apply. This allows us to pre-allocate memory to derive the exact memory usages, as well as per-group transfer and compute costs. Columns act like lines of code but more clearly express the details of axes and available optimization strategies than textual/symbolic methods. As polymorphic streamed algorithms are defined for the stream axis being of any size, we can begin the algorithm with a head $F$ taking an axis of size 0, initializing the maintained output to incorporate further information. As this expansion is built from the recursively expanded definition of a streamable function, correctness is ensured.

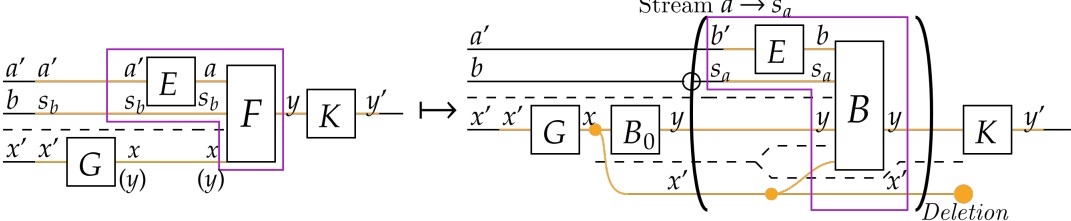

Figure 27: A streamed algorithm can be re-expressed with an explicit loop. The diagram illustrates how axes are partitioned, which variables are maintained, and what operations are applied iteratively.

By substituting the subalgorithms in Figure 27 with operations relevant to a particular algorithm, we can develop detailed pseudocode expansions for any streamed algorithm. This expansion is the first step in developing a hardware-aware algorithm. SoftMax-contraction, for example, can be expanded by substituting $B$ for the SoftMax-contraction accumulator from Appendix A.3.2, shown in Figure 28.

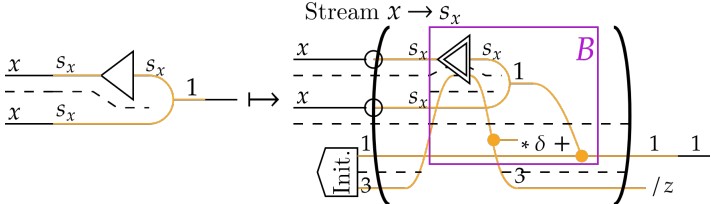

Figure 28: An example of pseudocode expansion. We re-express SoftMax-contraction using the accumulator derived in Appendix A.3.2.

Pseudocode expansion applied to attention involves substituting $E$ with query-key matrix multiplication, $F$ with SoftMax-contraction and, therefore, $B$ with the accumulator of SoftMax-contraction. We are also required to weave the top $\bar{q}$ (which is partitioned) and the bottom $d$ axes, deriving the two-level model pseudocode expansion for attention in a systematic manner. The outcome of this process is shown in Figure 29, which provides a two-level model expansion.

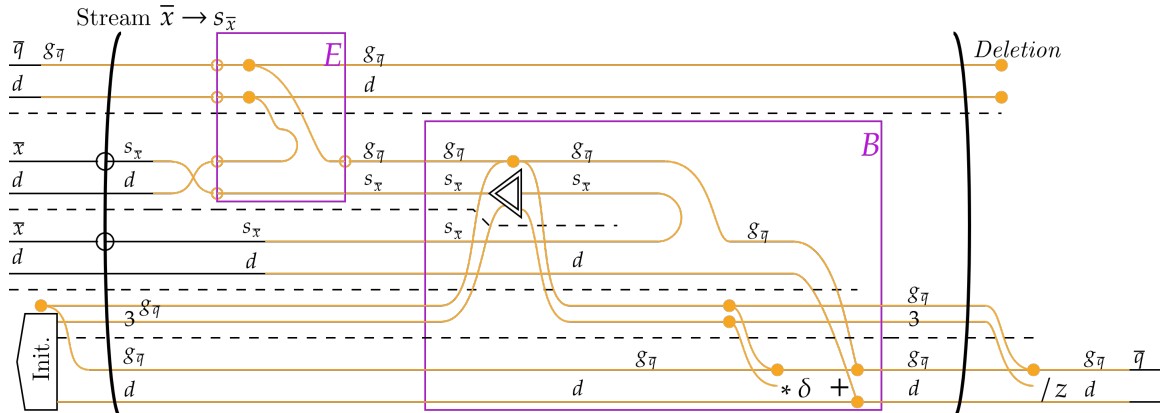

Figure 29: Step 1 of deriving a hardware-aware algorithm is expanding the two-level model into a pseudocode expression. We can apply the fusion theorems to attach $E$ and weave over the $\bar{q}$ and $d$ axes.

## 5.2 Subloops

Pseudocode expansions allow for streamable and split subalgorithms to be identified, which allow increased flexibility. Streamable subalgorithms are identified by inner parentheses (subloops). Note how accumulators are streamable, and therefore, the $B$ component of Figure 27 always allows for a subloop that streams $s_a$ along chunks of size $u_a$. Furthermore, matrix multiply-add operations can be split along any axis and linearly accumulated. We represent this by placing a dotted box around the operation. Shown in Figure 30, this is the second step in systematically developing a hardware-aware algorithm. At this point, we still have a hardware-independent two-level abstract representation of the algorithm.

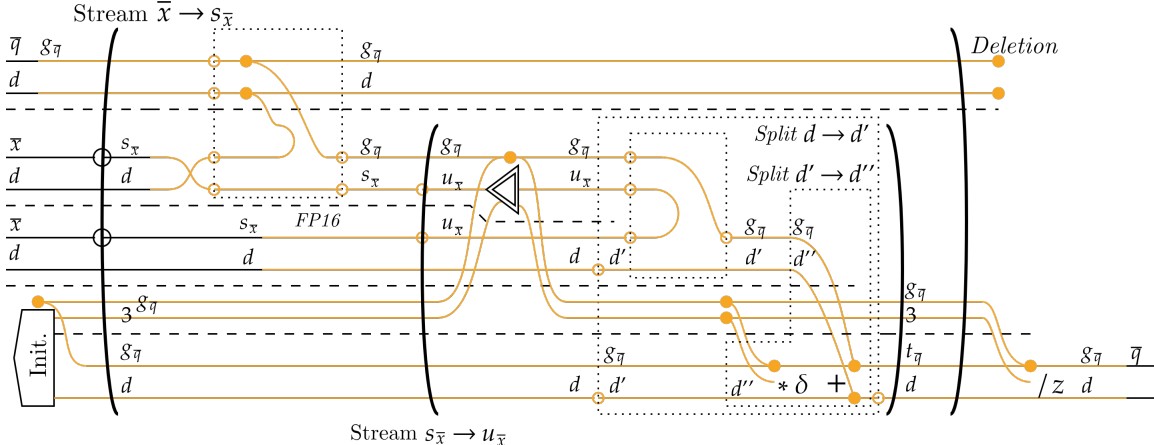

Figure 30: Step 2 requires identifying the subalgorithms and, hence, the degrees of freedom in axis sizes.

## 5.3 Recoloring Diagrams to Utilize Tensor Cores

We can impose hardware features by recoloring matrix multiplication and associated data as blue, and general-purpose operations as green. Tensor cores and thread levels are partitioned into groups within the $g_{\overline{q}}$ SMEM blocks, and we relabel $g_{\overline{q}}$ to $t_{\overline{q}}$ for threads and $w_{\overline{q}}$ for warpgroups. Transferring between these levels requires piping through orange SMEM. We also impose quantization labels. This generates Figure 31. At this point, the diagram only assumes the presence of tensor cores, and is therefore applicable to Volta, Ampere, Hopper, and similar non-NVIDIA architectures. Note that the second matrix-multiply add involves a scaling by $\delta$ weaved over the $g_{\overline{q}}$ axis. This is not possible with tensor cores, without a complex diagonal matrix approach which falls outside the standard paradigm of a systematic, diagrammatic derivation. This distinguishes our approach from FlashAttention.

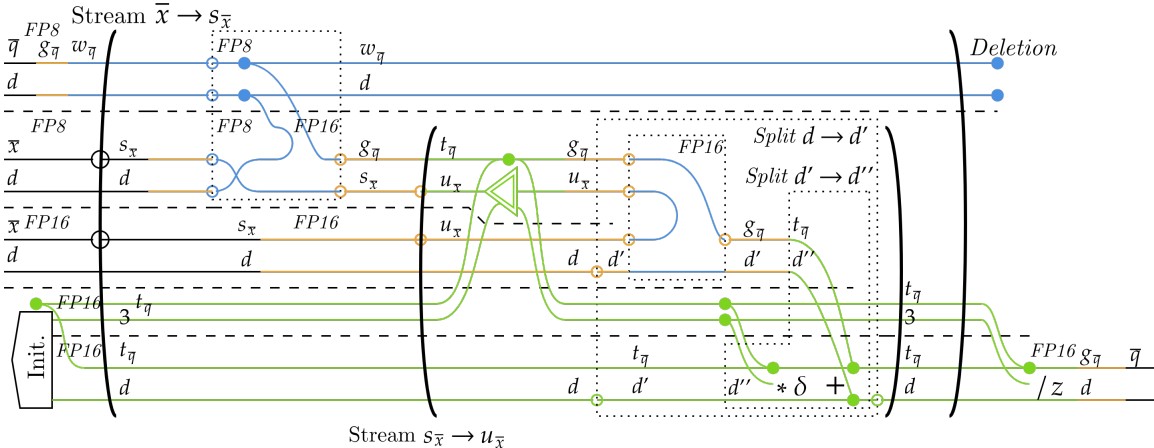

Figure 31: Step 3 involves assigning operations/data to specific cores and quantization levels.

## 5.4 Applying Divisor Constraints

Next, we impose the divisor constraints of specialized hardware features, including coalesced memory access and tensor cores. The divisor constraints come from the specific architecture being used. By imposing that $w_{\overline{q}}^{(128)}$ is divisible by 128, we indicate that we are working with warpgroups. We draw the axis sizes for tensor core operations from the specifications of Hopper.

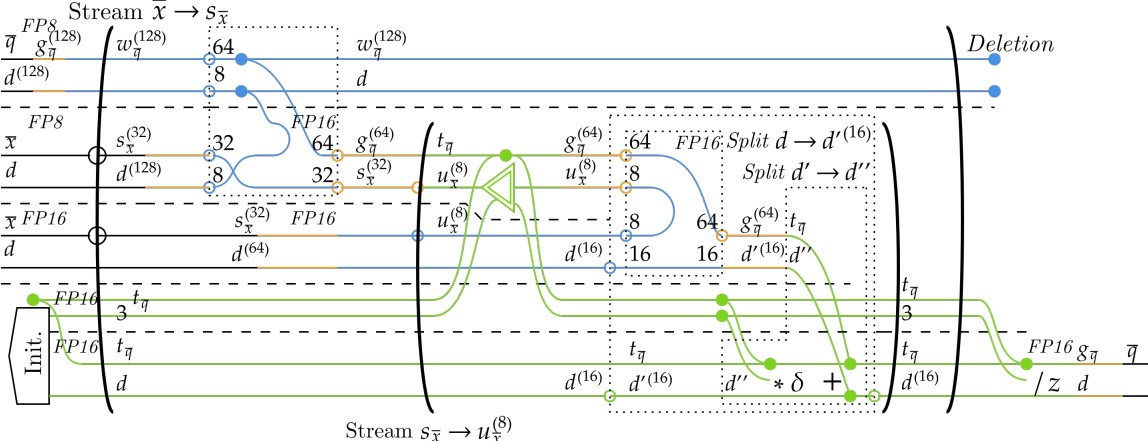

Figure 32: Step 4 has us impose divisor constraints, configuring the algorithm for a specific architecture. In this case, Hopper.

## 5.5 Identifying Variables

The data type-columns of diagrams indicate the location and size of variables simultaneously required in memory to execute the algorithm. Performance is best achieved by pre-allocating all required variables. From Figure 33, we identify all the required variable sizes. In some cases, we can reuse a variable at two points where it is not simultaneously required.

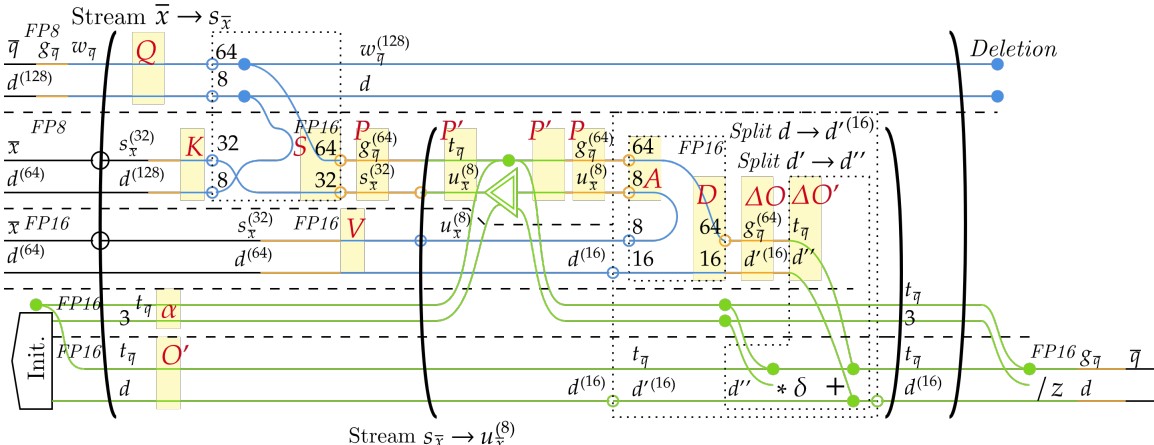

Figure 33: Step 5 identifies the required variables for accurate memory assessment and pre-allocation.

## 5.6 Configuration Table

Our aim is to find a configuration of axis sizes that maximizes performance while conforming to memory constraints. The memory limits are given by Hopper's specifications, which limits the SMEM and register memory per thread block. In Table 1, we list all the variables, the memory they use, and how they scale with the number of warpgroups, $N_{WG} = g_{\overline{q}}/128$. This allows us to find the maximum warpgroups per thread block given the configuration by $N_{\max} = (M_{\max} - M_{\mathrm{TB}})/M_{\mathrm{WG}}$ (see Table 2). In the attached document, we provide this table with tools to alter values (see Appendix B.2.1).

We configure the axis sizes in Table 1 to improve performance. Lower axis sizes allow for more warpgroups. This increases the degree of memory sharing, akin to increasing $g_{\overline{q}}$ in Figure 44. However, with $g_{\overline{q}} \geq 295$, we are not bandwidth bottlenecked on H100 (see Appendix B.0.1), even assuming no caching. Larger tiles

| | Variable | Size | Q. | Level | $M_{TB}$ (Bytes) | $M_{WG}$ (Bytes) |
|---|---|---|---|---|---|---|
| $Q$ | Queries | $w_{\bar{q}}^{(128)} \times d^{(128)}$ | FP8 | SMEM | | 16384 |
| $K$ | Keys | $s_{\bar{x}}^{(32)} \times d^{(128)}$ | FP8 | SMEM | 16384* | |
| $V$ | Values | $s_{\bar{x}}^{(32)} \times d^{(128)}$ | FP16 | SMEM | 32768* | |
| $S$ | Attention Scores | $w_{\bar{q}}^{(128)} \times s_{\bar{x}}^{(32)}$ | FP16 | Registers | | 16384 |
| $P$ | Weighted Scores | $g_{\bar{q}}^{(128)} \times s_{\bar{x}}^{(32)}$ | FP16 | SMEM | | 16384 |
| $P'$ | Weighted Scores (Registers) | $t_{\bar{q}} \times u_{\bar{x}}^{(8)}$ | FP16 | Registers | | 16384 |
| $A$ | Weighted Scores (Tensor Core) | $w_{\bar{q}}^{(128)} \times u_{\bar{x}}^{(8)}$ | FP16 | SMEM | | 8192 |
| $\alpha$ | Auxiliary | $t_{\bar{q}} \times 3$ | FP16 | Registers | | 768 |
| $O'$ | Output (Registers) | $t_{\bar{q}} \times d^{(128)}$ | FP16 | Registers | | 32768 |
| $D$ | Delta Output (Tensor Core) | $w_{\bar{q}}^{(128)} \times d'^{(16)}$ | FP16 | Registers | | 8192 |
| $\Delta O$ | Delta Output (SMEM) | $g_{\bar{q}}^{(128)} \times d'^{(16)}$ | FP16 | SMEM | | 8192 |
| $\Delta O'$ | Delta Output (Registers) | $t_{\bar{q}} \times d''$ | FP16 | Registers | | 2048 |
| | **Total (KB=1024B)** | | | SMEM | 48 | 48 |
| | **Total (KB)** | | | Registers | | 74.75 |

Table 1: The configuration table lists all the required variables and the memory they occupy. We set $w_{\bar{q}} = 128$, $t_{\bar{q}} = 1$, $s_{\bar{x}} = u_{\bar{x}} = 64$, $d = 128$, $d' = 32$, and $d'' = 8$. Alternative configurations can be explored in the linked document. *Keys and values are asynchronously loaded, doubling their memory usage.*

allow for greater parallelism and simplify the code, motivating the compiler to cooperate. The number of non-tensor FP16 multiply-add operations is dependent on the number of sub-loops, which we reduce to 1 by setting $u_{\bar{x}} = s_{\bar{x}}$. If we were working with FP16 queries and keys, we would have to aim for a more memory conservative configuration, possible decreasing the size of subloops. Configuration tables allow us to explore these possibilities in a systematic manner. In the associated document (Sheet "Configuration"), we include a file that allows configurations to be quickly assessed.

| | $M_{\max}$ | $M_{TB}$ | $M_{WG}$ | $N_{\max}$ | *Excess at $N=3$* | | |
|---|---|---|---|---|---|---|---|
| | | | | | Per TB. | Per WG. | Per thread (*Bytes*) |
| SMEM | 227 | 48 | 48 | 3.7 | 35 | 11.67 | |
| Registers | 256 | 0 | 74.75 | 3.4 | 31.75 | 10.58 | 85 |

Table 2: We find the maximum number of warpgroups given our configuration using a simple linear expression, $N_{\max} = (M_{\max} - M_{\text{TB}})/M_{\text{WG}}$. The excess registers per thread provides a buffer to prevent register spills which devastate performance.

## 5.7 Throughput Optimization

We can use diagrams to identify the bottlenecks of algorithms to guide optimized overlapping strategies. Each SM has specialized cores and pipelines for different operations which execute in parallel each clock cycle. GPUs will naturally implement a degree of overlapped execution, however, FlashAttention-3 (Shah et al., 2024) showed that explicit barriers can improve asynchrony.

From Figure 29, we find the ops per thread for each column. We take the baseline ops (eg. $2d$ for matrix multiplication) and multiply by the weaved axes. We ignore $g_{\bar{q}}$, to normalize for the number of active threads. In Table 3, we notate the pipeline each operation uses, the normalized ops, and the ops per clock cycle. We use these values to find the required clock cycle per column. These values are given per $s_{\bar{x}}$ subloop. The compute analysis is provided as Table 3 in the attached document for various algorithms (see Appendix B.2.1).

Table 3 indicates that clock cycles per iteration are lower bound by the 6 clock cycles per thread required for tensor cores. Waiting on any other actions will deviate performance from this lower bound. We can use

| Operation | $Q$-$K$ MatMul | SoftMax (*exponent*) | $P$-$V$ MatMul | FP16 Accumulate |
|---|---|---|---|---|
| Pipeline | Tensor | Special Function Unit | Tensor | FP16 |
| Ops/Th | $2ds_{\overline{x}} = 16384$ | $s_{\overline{x}} + s_{\overline{x}}/u_{\overline{x}} = 65$ | $2s_{\overline{x}}d = 16384$ | $2d(s_{\overline{x}}/u_{\overline{x}}) = 256$ |
| Ops/Clk | 8192 | 16 | 4096 | 512 |
| Clk/Th | 2 | 4.06 | 4 | 0.5 |

Table 3: We outline the clock cycles per thread required to execute different stages of the algorithm. The total clock cycles required are given by (Clk/Th)*(Number of Active Threads).

this lower bound to find the required warp groups to avoid being bandwidth limited. Assuming no caching, we find that $g_{\overline{q}}$ must be greater than 295. This is elaborated in Appendix B.0.1, where we also find the *ideal* tensor-core bottlenecked throughput to be 1.32 PFLOPs.

Hopper provides synchronization mechanisms between warp groups, which we exploit to shadow non-tensor core operations. Optimal performance is achieved by having barriers wait on tensor cores. We diagram a representation of the algorithm in Figure 34, with widths corresponding to required clock cycles per thread. We classify costs not considered in Table 3 as *overhead*. For example, type conversions are extremely slow, and exponential operations must occur in FP32. We represent overhead accommodation with hatched regions.

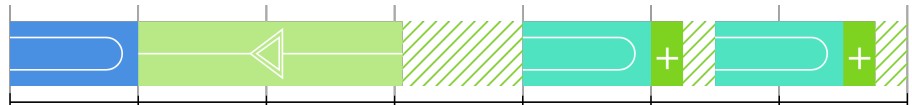

Figure 34: We can diagram the clock cycles of Figure 33 with a specific configuration ($u_{\overline{x}} = s_{\overline{x}}$, $d' = d/2$) by symbolic blocks, with width proportional to the clock cycles per thread. We use $d' = d/4$, but the strategy remains similar. Hatched areas indicate overhead, which we assume to be significant for non-tensor core operations. The ruler width is equal to the $Q$-$K$ FP8 tensor core matrix multiplication, and therefore represents 1 clock cycle per active thread.

We expand the diagram into three warpgroups. Each warpgroup acts independently, not relying on data from others, allowing us to shift operations forwards or backwards to wait on tensor cores. Furthermore, we can split the $Q$-$K$ tensor core matrix multiplication as it forms a linear subloop (see Figure 30). This allows us to construct a highly overlapped algorithm with two active iterations, shown in Figure 35. This algorithm will achieve optimal performance if the realized overhead is less than the accommodation sizes.

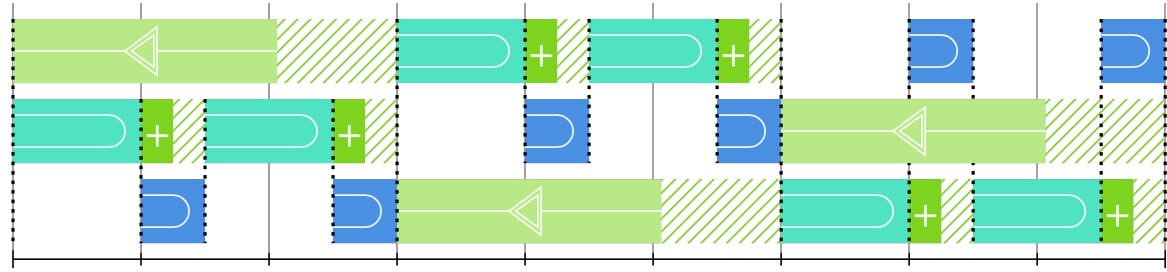

Figure 35: Three warpgroup pipelining strategy. Hatched areas indicate accommodation for overhead of non-tensor core operations. We have 50% overhead for SoftMax, and 100% for FP16 multiply-add. The ruler separates blocks of 128 clock cycles. Thick dotted lines indicate barriers. In the ideal case, these should all wait on tensor cores. We have two iterations active at once.

## 5.8 Conclusion of Derivation

In this section, we introduced a systematic method for deriving, configuring, and analyzing algorithms for our toy Hopper hierarchy. The utility of this approach is in quickly deriving sketches of optimized algorithms with a clear idea of expected performance and how the algorithm can fall short. The diagrammatic methodology can be extended to simulate additional architectures for which kernels have not yet been developed, including new releases like Blackwell (NVIDIA, 2025) and non-NVIDIA hardware (AMD, 2023).

The toy model, however, overlooks certain features of GPU algorithms. Though diagrams provide accommodation for overhead, we have ignored the overhead of tensor core operations and assumed that latency is completely hidden. In practice, latency may delay algorithm execution, and tensor core operations may incur overhead from small tile sizes (Bikshandi & Shah, 2023). The aim of this work is to develop a theoretical framework that allows specific assumptions (like tensor core overhead) to be improved by empirical testing. Rather than generally stating "larger tensor core tile sizes improve performance", we may be able to claim that "the overhead of tensor core operations is $Y$ given a tile size $X$".

A major concern is our lack of utilization of tensor core-fragmented register-level operations. We have modeled tensor core memory as incoherent, however, methods exist to find how fragments are stored across registers, which can be exploited to avoid intermediate communications with SMEM (Bikshandi et al., 2023). This requires special treatment, as we would have to consider *reindexing* operations which manipulate how data is accessed without changing values. This can be encompassed by a reindexing weaves (Abbott, 2023), which can capture the implementation details of CUTLASS layouts and swizzled memory (Shah, 2024).

## 6 Comparison to FlashAttention

Our diagrammatic approach and "ideal" algorithm provide insight into FlashAttention-3. The key difference in the operations employed is that FlashAttention-3 manipulates tensor core data thread-wise. This allows it to use less variables than our approach, as we document in Appendix B.2. This functionality is not currently provided by diagrams, meaning our algorithm resorts to using SMEM data. Properly considering the manipulation of tensor core data would require investigating indexing with diagrams in-depth. An ad-hoc treatment would detract from the emphasis on systematic methods of this work. However, the value of diagrams lies in the systematic methodology and analytical tools, which we can be applied to better understand FlashAttention.

### 6.1 FP16 FlashAttention-3

FP16 FlashAttention-3 uses FP32 auxiliary variables and intra-warp group overlapping. We outline its configuration in Appendix B.2.4, linking to the attached document. Large tile sizes are used, with effective $g_{\overline{q}} = 128$ and $s_{\overline{x}} = 128$. Intra-warp group overlapping requires a single warpgroup to hold $S^{\text{next}}$, overloading the limited number of per register threads, causing spillage[1].

We diagram the intra-warpgroup strategy and a hypothetical alternative inter-warpgroup approach for $d = 128$ in Figure 36. We see that the intra-warpgroup approach does not accommodate overhead for SoftMax, potentially delaying the algorithm. The inter-warpgroup strategy provides 100% overhead for SoftMax. This invites empirical testing where we reduce the tile size to fit multiple warpgroups per SM. If an intra-warpgroup strategy remains superior, we can update the assumptions of our toy hierarchy based on the empirical results.

The choice of a large $s_{\overline{x}}$ is the result of empirical testing and a trade-off between register pressure and the performance benefits of large tiles. This indicates our model can be improved by considering the overhead of tensor core operations, especially for small tile sizes. Without caching, the algorithm will be bandwidth-bound. Nonetheless, the algorithm is performant, achieving 740 of the maximum 989 TFLOPs of compute (75%). Diagrams like Figure 32 show quantization changes more clearly than code, and we note that it is likely the case that much of the value of FP32 accumulators is lost in the second matrix multiplication.

---

[1]Exceeding register limits causes *local memory* to be used. Despite its name, it is located off the SM.

FP16 Inter-Warpgroup Overlapping

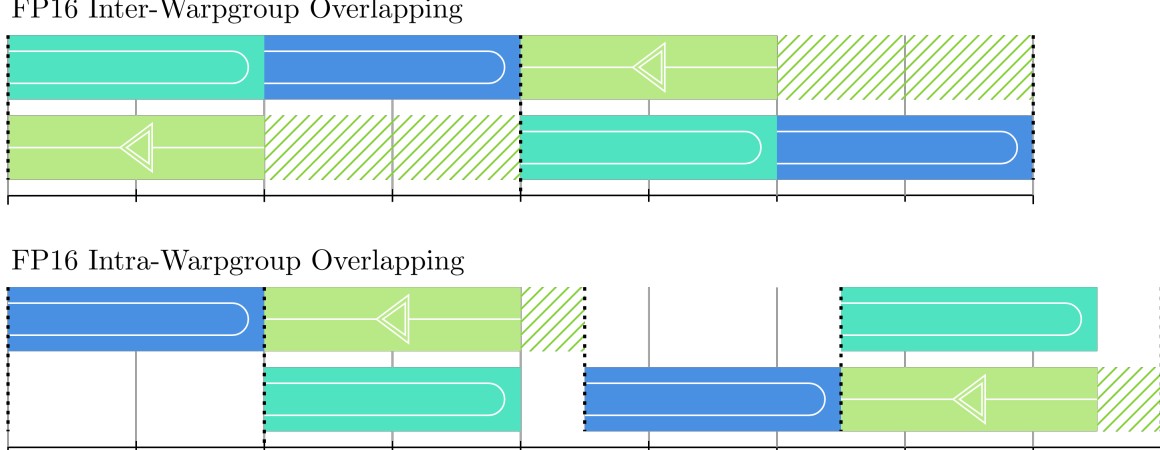

FP16 Intra-Warpgroup Overlapping

Figure 36: FP16 inter and intra warpgroup overlapping strategies. Hatched regions indicate SoftMax overhead.

## 6.2 FP8 FlashAttention-3

FP8 FlashAttention-3 uses FP16 auxiliary variables and inter-warpgroup overlapping in the original version, documented in Appendix B.2.2. However, newer versions seem to use the intra-warpgroup overlapping operation (see Appendix B.2.3). This strategy allows a single warpgroup to store a large $s_{\overline{x}}$ value, which seems to have an immense benefit. FP8 FlashAttention-3 is bound by the slow special function unit required for the SoftMax function. In empirical testing, it achieves 1.2 PFLOPs of the maximum 1.98 (60% utilization).

In Figure 37, we diagram the overlapping strategies for FP8. We use our clock cycle assessments to show overlapping operations and assess the impact of function overhead. We provide diagrams for both an inter-warpgroup strategy (original version) and an intra-warpgroup approach (mimicking the GitHub). The clock cycle analysis shows that the strategies are both delayed by the overhead of the SoftMax operation. As the SoftMax operation utilizes as many clock cycles as tensor core operations, this can not be improved upon. Delays from unit conversions and other factors will delay the algorithm, reducing the maximum achievable FLOPs. Furthermore, the FP8 intra-warpgroup strategy has the special function unit (the limiting factor) inactive for much of the cycle, reducing performance by a minimum of 33%, which explains much of the underutilization.

FP8 Inter-Warpgroup Overlapping

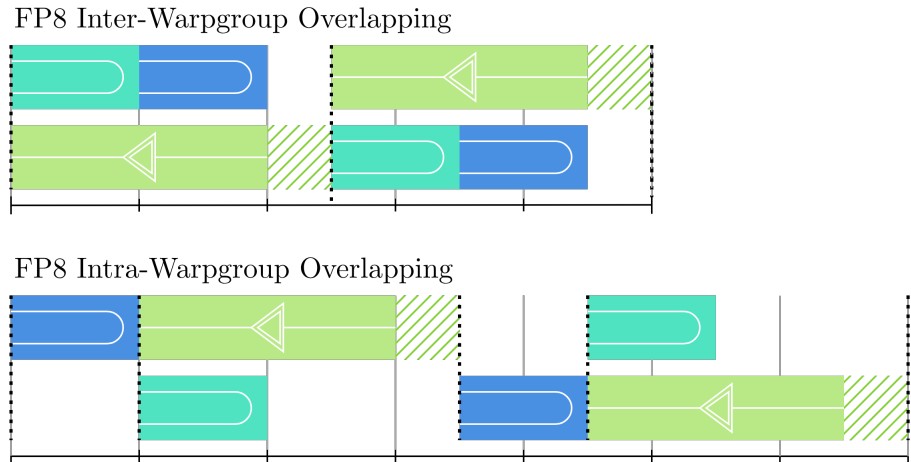

FP8 Intra-Warpgroup Overlapping

Figure 37: FP8 is bottlenecked by the SoftMax operation, and therefore has no accommodation for its overhead, no matter the overlapping strategy employed.

### 6.3 Further Comments

The aim of this analysis is to see how our assumptions and predictions can be compared to a prior successful kernel. The choices made for FlashAttention-3, especially in the latest releases, of intra-warpgroup strategies, large $s_{\overline{x}}$ tiles, and a small number of warpgroups $g_{\overline{q}}$, suggests that there is a significant benefit to large inner dimensions for tensor core operations. If this is the case, our model would need to be updated. This does not detract from this work's main goal — we aim to propose a theoretical framework that can be updated and tested in exactly this manner. In the long term, the exact details are secondary to the mechanism that allows them to be updated. By positing certain assumptions about constructing an optimal algorithm as in Section 5, and seeing how it falls short, we can inform future development.

Our model predicts that FlashAttention-3 is bottlenecked by the SoftMax operation. Interestingly, if we assume a 66% overhead in SoftMax, we would expect FP16 intra-warpgroup to take 4/3 as long as necessary and FP8 inter warpgroup to take 5/3 as long as necessary, explaining both the 75% utilization for FP16 and the 60% utilization for FP8 in the original paper. This is an alternative hypothesis from delays originating from the overhead of small tile sizes, and invites empirical testing.

## 7 Conclusion and Future Work

In this work, we have contributed a diagrammatic method for representing deep learning algorithms. In Section 3, we showed how this allows for the rapid derivation of high-level optimization strategies, with an in-depth associated performance model shown in Section 4. In Section 5, we used this framework to develop a systematic methodology for deriving hardware-aware algorithm. In Section 6, we use our methodology to better understand FlashAttention, making testable claims about the factors that limit performance.

This work is a zero-to-one attempt at creating a systematic approach to understanding IO-Awareness, and invites a scientific approach to GPU optimizations. Diagrams rely on assumptions about GPU behaviour to predict real-world performance. By separating theory from empirical results, we are able to treat these assumptions as scientific hypotheses. Experiments can be run, with their failure or success guiding improvements in the framework. The diagrammatic framework can therefore avoid post-hoc rationalization. Furthermore, diagrams reveal the many possible configurations of algorithms as in Figure 30 and 32. These configurations can act as templates over which a host of experiments can be run. As the framework becomes increasingly capable of predicting performance, the cost of empirically testing various configurations can decrease.

This work focuses on attention. However, diagrams can provide value by extending to other algorithms. Mixture-of-expert models (Mistral AI team, 2024) use immense resources, making optimizations particularly impactful. Additional strategies can be formalized. Convolution and sliding window attention (Beltagy et al., 2020) reindex weavings (Abbott, 2023), operations which change *how* data is accessed without changing the data itself. Reindex weavings can integrate CUTLASS' tensor storage system, linking diagrams to CUTLASS implementations and providing access to tensor core-level weaved operations.

Diagrams conform to a category-theoretic description, which is not covered in this paper. A categorical perspective would have diagrams encoded as a formal syntax (Piedeleu & Zanasi, 2025) which opens up automated compilation tools (Wilson, 2023), consideration of backpropagation (Fong et al., 2019; Cruttwell et al., 2022), and estimating error accumulation (Perrone, 2024). A formal treatment would streamline our theorems, and provide access to tools that allow additional strategies to be considered. Convolution and sliding window attention (Beltagy et al., 2020) use reindexings on weavings. These operations change *how* data is accessed without changing the data itself, and correspond to Yoneda natural transformations from category theory (Abbott, 2023). Reindexings would allow CUTLASS' tensor storage system to be incorporated into diagrams, and introduce a systematic method for optimizing convolution-like algorithms. Furthermore, we can develop methods to distribute streams across cores and use accumulators as multi-core reduction operations (Osama et al., 2023). Accumulators-as-reductions requires special treatment of transfer costs, but would allow wave quantization to be considered which is outside the scope of this paper.

Furthermore, we can use our framework to develop a systematic approach to optimization that considers hardware design, capabilities, and algorithm configuration to performance. This can be constructed by adapting the framework of categorical co-design (Zardini, 2023), which relates functionalities to requirements in a compositional manner. Given the categorical nature of diagrams, this would the development of a holistic approach across the engineering stack of optimizing deep learning deployment, relating hardware, algorithms, and configuration choices towards optimal performance and minimal resource usage.

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

# A  Appendix

## A.1  Fusion Theorem

**Theorem 1 (Fusion Theorem)** *Composition and weaving of a streamable algorithm which does not remove the streamed axis yields a streamable algorithm.*

Streamable algorithms satisfies the form in Figure 38, allowing the recursive expansion of Figure 39. Streamability depends on polymorphism over the $a$ axis, meaning the function/algorithm is defined for the axis being of any size, and the existence of an accumulator $B$, which allows the streamed input data to be split.

Figure 38: A streamable algorithm requires an accumulator $B$ which allows the polymorphic streamable axis to be split. We can fuse the algorithm with a head and tail, which do not require additional loads and saves if their memory usage is sufficiently small.

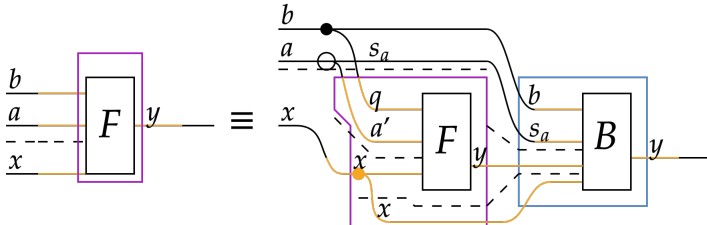

This allows for recursive decomposition, reducing the size of the remainder $a'$ axis to $a' \leq s_a$ as in Figure 39. Here, we add a head and a tail, which are not expanded but can be executed without an additional transfer and are hence fused. This requires their memory usage to be sufficiently small to not exceed hardware limitations. Composition on $G$ or $K$ simply replaces those algorithms with the composed form, yielding a modified head/tail for the streamable algorithm.

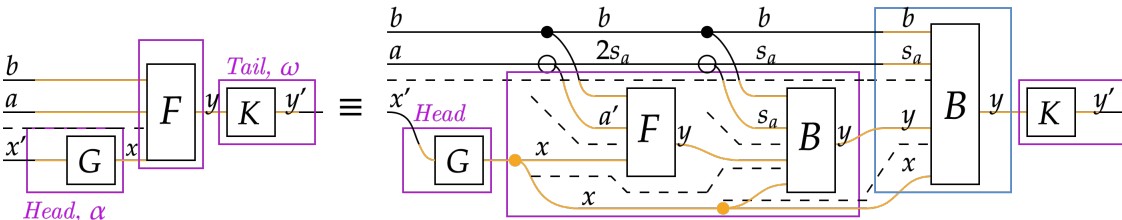

Figure 39: If Figure 38 is satisfied, then the algorithm can be recursively decomposed. This reduces the size of chunks until the on-chip memory is sufficiently small.

Composing by $E$ on the $b$-axis with an algorithm weaved by the streamable $s_a$ axis, we can exploit a partition copy (see Figure 10) to show that the composed $F'$ has an accumulator $B'$.

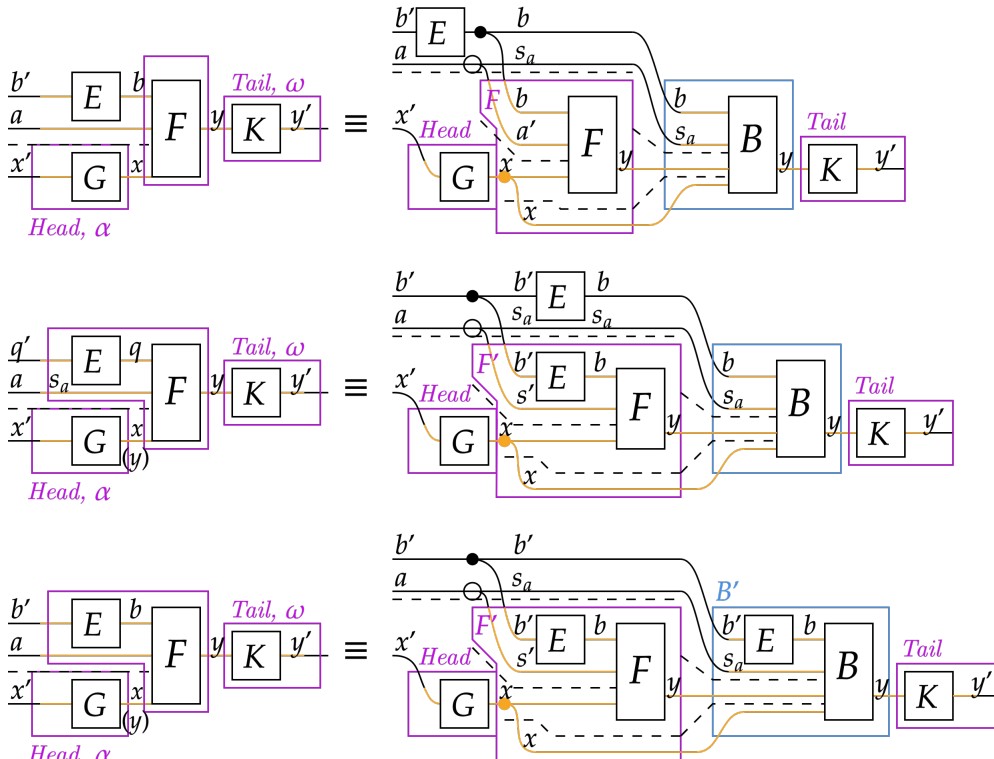

Finally, we are required to show that weaving preserves streamability. This exploits a characteristic of mapping composed functions. Mapping a composed function over an additional axis is equivalent to composing

the individual functions mapped over that axis. Therefore, we can show that a weaved $B$ is an accumulator for a weaved $F$ as in Figure 40.

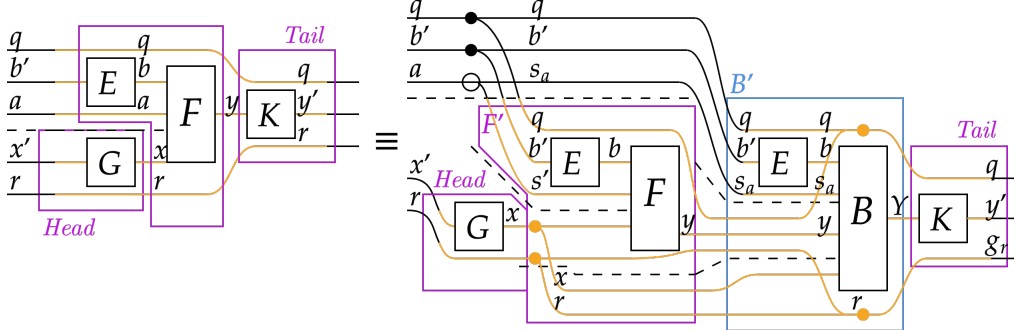

Figure 40: Weaving over $F$ composed with $E$ and $G$ at its inputs is equivalent to weaving over the smaller $F$ and $B$ algorithms which, when composed, give $F$. This weaves need to trace over the inputs which they target.

We can combine all the above expressions into the single form of Figure 41, where we also apply group partitioning. This separates the mapped axis into groups of size $g_q$ distributed across processors. We can iteratively apply these rules; an algorithm can be composed with an algorithm $E$ weaved over the streamable axis. This generates a streamable algorithm, which can be weaved over one of the inputs introduced by $E$. This is used to construct streamable (flash) attention from a SoftMax-Contraction kernel.

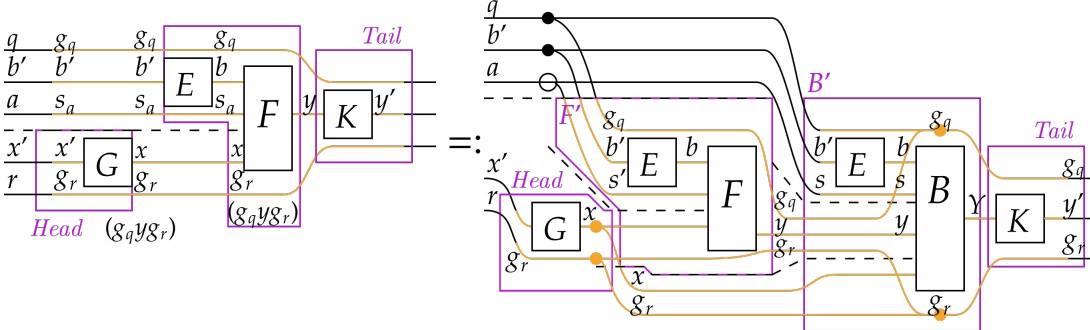

Figure 41: We can combine the streaming theorems into a single expression. Given that the algorithm $F$ is streamable along the $a/s_a$ axis, we can add modifications to generate a new streamable algorithm. We can group partition along the streamed axes, mapping groups of size $g_q \times g_r$ to different processors.

## A.2 Multi-Level Performance Models

We define for a two-level model diagram the optimal transfers $H^*(\vec{a}, M)$, where $\vec{a}$ are the axis sizes and $M$ is the maximum memory available at the lower level. We use $\vec{g}$ to indicate some configuration of group sizes. We are interested in memory usage in the limit of large $M$ and $\vec{g}$, in which case the limiting factor is some $y \, \Pi_i \, g_i$ weaved by all grouped axes, with $y$ typically being the output. For smaller $M$, we aim for a specific configuration as in Section 5. Two-level diagrams derived optimal transfers solve for Equation 4, where $i$ iterates over the grouped axes:

$$H^*(\vec{a}, M) = \min_{\vec{g}} \, \frac{\Pi_i \, a_i}{\Pi_i \, g_i} \, H_g(\vec{a}, \vec{g}) \text{ given } M \geqslant y \, \Pi_i \, g_i \tag{4}$$

We extend this to multiple levels be assigning a memory $M_\ell$ to each level and a weighted transfer cost $\dot{H}_\ell^{-1}$, which represents bandwidth. As described in Section 4, this conforms to;

$$H^* = \sum_\ell \dot{H}_\ell^{-1} \ H^*(\vec{a}, M_\ell) \tag{5}$$

How additional levels are used, either as intermediate caches or cross-transfer levels, use the same general performance model of Equation 5 but with altered weighted transfer costs $\dot{H}_\ell^{-1}$ and per-level effective memory $M_\ell$. Using $M_\ell^{\max}$ to indicate the maximum memory per level, $N_\ell^{\max}$ to indicate the maximum number of child nodes per higher level node, $h$ for the immediately higher level and $c$ for the immediately lower, and $\dot{H}_{\ell \to \ell'}^{-1}$ as the raw weighted transfer cost between $\ell$ and $\ell'$, we adapt our performance model to get;

$$\dot{H}_\ell^{-1} = \begin{cases} \dot{H}_{h \to c}^{-1} - \dot{H}_{\ell \to c}^{-1} & \ell \text{ is a cross-transfer level} \\ \dot{H}_{h \to \ell}^{-1} & \text{else} \end{cases}$$

$$M_\ell = \begin{cases} N_c^{\max} M_c & \ell \text{ is a cross-transfer or intermediate cache} \\ M_\ell^{\max} & \text{else} \end{cases}$$

Therefore, using cross-transfer or caching levels conforms to our standard performance models. This lets Equation 2 act as a universal performance model for multi-level GPUs, and ensures that $\dot{H}_\ell^{-1} M_\ell^{-\beta}$ is the characteristic property for comparing the performance of algorithms for different low-level memory sizes.

### A.2.1 Intermediate Caching

**Theorem 2** *An intermediate caching level $\ell 1$ for an output restricted algorithm with a number restriction $N_{\ell 2}^{\max} \geqslant N_{g,\ell 2}/N_{g,\ell 1}$ conforms to the standard performance model with $M_{\ell 1} = M_{\ell 2} N_{\ell 2}^{\max}$, requiring $H^*(\vec{a}, M_{\ell 2} N_{\ell 2}^{\max})$ total transfers.*

For an intermediate caching model which is output limited, we have the standard constraint for the lower level, $M_{\ell 2} \geqslant y \, \Pi_i \, g_{\ell 2, i}$, and the number restriction, $N_{\ell 2}^{\max} \geqslant N_{g,\ell 2}/N_{g,\ell 1} = \Pi_i \, g_{\ell 1, i}/\Pi_i \, g_{\ell 2, i}$. We aim to find the configuration of $\vec{g}_{\ell 1}$ and $\vec{g}_{\ell 2}$ which minimizes the number of transferred values. Assuming the algorithm is output limited, the effective size of the caching column is $y \, \Pi_i \, g_i^{\ell 1}$. This lets us express the restrictions as:

$$H_{\ell 1}^* = \min_{\vec{g}} \frac{\Pi_i \, a_i}{\Pi_i \, g_{\ell 1, i}} \ H_g(\vec{a}, \vec{g}_{\ell 1}) \qquad\qquad \text{given } N_{\ell 2}^{\max} \geqslant \Pi_i \, g_{\ell 1, i}/\Pi_i \, g_{\ell 2, i} \tag{6}$$

$$H_{\ell 2}^* = \min_{\vec{g}} \frac{\Pi_i \, a_i}{\Pi_i \, g_{\ell 2, i}} \ H_g(\vec{a}, \vec{g}_{\ell 2}) \qquad\qquad \text{given } M_{\ell 2} \geqslant y \, \Pi_i \, g_{\ell 2, i} \tag{7}$$

We can multiply the restriction of (6) by the restriction of (7) to get (8). Assuming that the inequalities are sufficiently close to equalities, we outline the new problem to solve:

$$H_{\ell 1}^* = \min_{\vec{g}} \frac{\Pi_i \, a_i}{\Pi_i \, g_{\ell 1, i}} \ H_g(\vec{a}, \vec{g}_{\ell 1}) \qquad\qquad \text{given } M_{\ell 2} N_{\ell 2}^{\max} \geqslant y \, \Pi_i \, g_{\ell 1.i} \tag{8}$$

$$H_{\ell 2}^* = \min_{\vec{g}} \frac{\Pi_i \, a_i}{\Pi_i \, g_{\ell 2, i}} \ H_g(\vec{a}, \vec{g}_{\ell 2}) \qquad\qquad \text{given } M_{\ell 2} \geqslant y \, \Pi_i \, g_i^{\ell 2} \tag{9}$$

These restrictions correspond to Equation 4, so we can substitute in $H^*(\vec{a}, M_\ell)$ for both levels, but using $M_{\ell 2} N_{\ell 2}^{\max}$ for the intermediate level instead of its own maximum memory, $M_{\ell 1}^{\max}$. We therefore have;

$$H_{\ell 1}^* = H^*(\vec{a}, M_{\ell 2} N_{\ell 2}^{\max})$$
$$H_{\ell 2}^* = H^*(\vec{a}, M_{\ell 2})$$

A caching level therefore conforms to our standard multi-level performance model derived from a two-level diagram, but with the intermediate level using total lower level memory instead of its own.

### A.2.2   Cross-Transfer Level

**Theorem 3** *Introducing a cross-transfer level $x$ between a higher level $h$ and a lower level $c$ allows us to replace the weighted transfer cost of an output-limited algorithm;*

$$H^* = \dots \; + \dot{H}_{h \to c}^{-1} H^*(\vec{a}, M_c) + \; \dots$$

*With,*

$$H^* = \dots \; + \; (\dot{H}_{h \to c}^{-1} - \dot{H}_{x \to c}^{-1}) \; H^*(\vec{a}, N_c^{\max} M_c) + \dot{H}_{x \to c}^{-1} H^*(\vec{a}, M_c) + \; \dots$$

*Where $\dot{H}_{h \to c}^{-1}$ is the weighted transfer cost of sending data to children, and $\dot{H}_{x \to c}^{-1}$ is the weighted transfer cost of sending data between children.*

The child level $c$ requires a total of $H^*(\vec{a}, M_c)$ data transfers from the higher level, typically incurring a weighted transfer cost per value of $\dot{H}_{h \to c}^{-1}$. With a cross-transfer level $x$ between $h$ and $c$, we can send some data $H_x^*$ to any of the children and cross-transfer the remaining at a weighted transfer cost of $\dot{H}_{x \to c}^{-1}$ per value. We need to derive the optimal configuration of $\vec{g}_x$ to minimize $H_x^*$. This configuration incurs a number restriction, as the child processors need to remain active. We therefore have,

$$H_x^* = \min_{\vec{g}} \frac{\Pi_i \; a_i}{\Pi_i \; g_{x,i}} \; H_g(\vec{a}, \vec{g}_x) \qquad\qquad \text{given } N_c^{\max} \geqslant \Pi_i \; g_{x,i}/\Pi_i \; g_{c,i}$$

$$H_c^* = \min_{\vec{g}} \frac{\Pi_i \; a_i}{\Pi_i \; g_{c,i}} \; H_g(\vec{a}, \vec{g}_c) \qquad\qquad \text{given } M_c \geqslant y \; \Pi_i \; g_{c,i}$$

We perform a similar substitution to Section A.2.1, yielding a new set of restrictions;

$$H_x^* = \min_{\vec{g}} \frac{\Pi_i \; a_i}{\Pi_i \; g_{x,i}} \; H_g(\vec{a}, \vec{g}_x) \qquad\qquad \text{given } M_c N_c^{\max} \geqslant y \; \Pi_i \; g_{x,i}$$

$$H_c^* = \min_{\vec{g}} \frac{\Pi_i \; a_i}{\Pi_i \; g_{c,i}} \; H_g(\vec{a}, \vec{g}_c) \qquad\qquad \text{given } M_c \geqslant y \; \Pi_i \; g_{c,i}$$

These restrictions conform to the two-level model optimal, with required transfers being $H_x^* = H^*(\vec{a}, M_c N_c^{\max})$ and $H_c^* = H^*(\vec{a}, M_c)$. When considering transfers for the total weighted transfer cost calculation, we can subtract the required transfers to $x$ from the transfers required to $c$. This is because data is transferred from the higher level to the cross-transfer level by sending it to children, so much of the data is already available. This results in the substituting the total weighted transfer costs (10) with (11):

$$H^* = \dots \; + \dot{H}_{h \to c}^{-1} H^*(\vec{a}, M_c) + \; \dots \tag{10}$$

$$\mapsto H^* = \dots \; + \dot{H}_{h \to c}^{-1} H^*(\vec{a}, N_c^{\max} M_c) + \dot{H}_{x \to c}^{-1} \left( H^*(\vec{a}, M_c) - H^*(\vec{a}, N_c^{\max} M_c) \right) + \; \dots \tag{11}$$

We can rearrange (11) so that we have the standard format of each level having $H^*(\vec{a}, M_\ell)$ transfers by instead modifying the transfer cost:

$$H^* = \dots \; + \left( \dot{H}_{h \to c}^{-1} - \dot{H}_{x \to c}^{-1} \right) H^*(\vec{a}, N_c^{\max} M_c) + \dot{H}_{x \to c}^{-1} \; H^*(\vec{a}, M_c) + \; \dots \tag{12}$$

Therefore, a cross-transfer level conforms to the standard performance model. Instead of using the weighted transfer cost of sending data to the children $\dot{H}_{h \to c}^{-1}$ for the cross-transfer level, we set $\dot{H}_x^{-1} = \dot{H}_{h \to c}^{-1} - \dot{H}_{x \to c}^{-1}$ and we remap $\dot{H}_c = \dot{H}_{x \to c}^{-1}$. This lets us express (12) as the expression below, which conforms to the standard multi-level performance model of (5):

$$H^* = \dots \; + \dot{H}_x^{-1} \; H^*(\vec{a}, N_c^{\max} M_c) + \dot{H}_c^{-1} \; H^*(\vec{a}, M_c) + \; \dots$$

### A.2.3   Additional Notes

In the case of a multi-GPU hierarchy, the interconnected topology is a cross-transfer level $x$ which distributes data among child GPUs $c$ at a weighted transfer cost of $\dot{H}_{x \to c}^{-1}$. If we assume data is already distributed across GPUs, then the number of GPU cross-transfers is $H^*(\vec{a}, M_c^{\max}) - H^*(\vec{a}, N_c^{\max} M_c^{\max})$. So far, we

have considered the highest level to have unlimited memory and zero weighted transfer cost. We can model multi-GPU systems by the highest level (the multi-GPU level $x$) as having a memory equal to $N_c^{\max} M_c^{\max}$ and negative weighted transfer cost $-\dot{H}_{x \to c}^{-1}$, which assumes data is already distributed among GPUs. This provides a rough model, which can be refined by aligning the group sizes used in diagrams.

Often, we can configure the number of cross-transfer level children to alter the cross-transfer bandwidth. In (Luo et al., 2024), it was found that the bandwidth of H800 (*Chinese market Hopper GPUs*) SM-SM transfers varies from $3.27TB/s$ with a cluster size $N = 2$ to $2.65TB/s$ with a cluster size of $N = 4$, compared to $2.04TB/s$ of GMEM-SMEM bandwidth. This imposes a trade-off; smaller cluster sizes improve effective $\dot{H}_c$ but reduce the cross-transfer discount $H^*(\vec{a}, N_c^{\max} M_c^{\max})$. (Luo et al., 2024) notes that balancing this trade-off is an important direction for exploration. We can use our model to find the difference in weighted transfer costs with (11) and without (10) a cross-transfer level, providing an equation to optimize for $N$.

$$
\begin{aligned}
\Delta H^* &= \left( \ldots + \dot{H}_{h \to c}^{-1} H^*(\vec{a}, M_c) + \ldots \right) \\
&\quad - \left( \ldots + \left( \dot{H}_{h \to c}^{-1} - \dot{H}_{x \to c}^{-1} \right) H^*(\vec{a}, N\ M_c) + \dot{H}_{x \to c}^{-1}\ H^*(\vec{a}, M_c) + \ldots \right) \\
&= \left( \dot{H}_{h \to c}^{-1} - \dot{H}_{x \to c}^{-1} \right) H^*(\vec{a}, M_c) - \left( \dot{H}_{h \to c}^{-1} - \dot{H}_{x \to c}^{-1} \right) H^*(\vec{a}, N\ M_c) \\
&= \left( \dot{H}_{h \to c}^{-1} - \dot{H}_{x \to c}^{-1} \right) \left( H^*(\vec{a}, M_c) - H^*(\vec{a}, N\ M_c) \right) \\
\Delta H^* &= \Delta \dot{H}^{-1}(N) \sum_t \alpha_t(\vec{a})\ M_c^{-\beta} \left( 1 - N^{-\beta} \right)
\end{aligned}
$$

### A.3 Streamability

#### A.3.1 Contraction

The streamability of contraction (a dot product) requires that an accumulator exists of the form in Figure 38. Contraction for vectors $v, w \in \mathbb{R}^a$ is given by $v \cdot w = \Sigma_{i=0}^{n-1} v_i \cdot w_i$, which can be expressed as $v \cdot w = \Sigma_{i=0}^{s'-1} v_i \cdot w_i + \underline{\Sigma_{i=s'}^{a-1} v_i \cdot w_i}$, where the underlined portion is the accumulator. We diagrammatically show this in Figure 42, highlighting the accumulator in blue.

Figure 42: We can re-express contraction as an initial function followed by the accumulator. Any size can be chosen for $s$ and $s'$, and the expression can be recursively expanded until $s' \leq s_a$ for some target stream batch size $s_a$.

#### A.3.2 Fusion of SoftMax-Contraction

SoftMax-Contraction is streamable by maintaining a running maximum and sum on chip as auxiliary variables. We express this by streaming unscaled SoftMax-Contraction with the initialization of the auxiliary variables as the head and scaling by the sum as a tail as in Figure 43.

Figure 43: Streamable SoftMax-Contraction is implemented by accumulating the results of unscaled SoftMax-Contraction applied to segments, adjusting the baseline relative to the current maximum value. We apply the normalization by $z$ as a tail for the expression.

We can derive streamable SoftMax-Contraction by taking recursively expanded Auxiliary SoftMax (Figure 44) and contracting its output. Recursively expanded Auxiliary SoftMax is not streamable as the memory usage increases with the input size. However, it terminates in a join, allowing it to be fused with a contraction.

Figure 44: Auxiliary SoftMax (defined in Table 4), where we maintain auxiliary variables, can be recursively expanded.

| Base SoftMax | Auxiliary SoftMax | Unscaled SoftMax |
|---|---|---|
| | | |
| SoftMax$(\vec{x})$ : 
 $\quad \mu \leftarrow \max(\beta x_i)$ 
 $\quad s_i \leftarrow \exp(\beta x_i - \mu)$ 
 $\quad z \leftarrow \Sigma_i\ s_i$ 
 $\quad y_i \leftarrow s_i/z$ 
 $\quad$ Return $\vec{y}$ | Initialize() : 
 $\quad$ Return $(-\infty, 0, 0)$ 
 SoftMax$_0(\vec{x}, (\mu', \delta z', z'))$ : 
 $\quad \mu \leftarrow \max(\beta x_i, \mu')$ 
 $\quad s_i \leftarrow \exp(\beta x_i - \mu)$ 
 $\quad \delta \leftarrow \exp(\mu' - \mu)$ 
 $\quad z \leftarrow \delta z' + \Sigma_i\ s_i$ 
 $\quad y_i \leftarrow s_i/z$ 
 $\quad$ Return $\vec{y}, (\mu, \delta z', z)$ 

 *Scaling (on prior values),* 
 $\vec{y}' \mathrel{*}= \vec{y}'\ \delta z'/z$ | Initialize() : 
 $\quad$ Return $(-\infty, 0, 0)$ 
 SoftMax$_1(\vec{s}, (\mu', \delta', z'))$ : 
 $\quad \mu \leftarrow \max(\beta x_i, \mu')$ 
 $\quad s_i \leftarrow \exp(\beta x_i - \mu)$ 
 $\quad \delta \leftarrow \exp(\mu' - \mu)$ 
 $\quad z \leftarrow \delta\ *\ z' + \Sigma_i\ s_i$ 
 $\quad$ Return $\vec{s}, (\mu, \delta, z)$ 
 *Scaling (on prior values),* 
 $\vec{y}' \mathrel{*}= \vec{y}'\ \delta$ 
 *Scaling (at tail),* 
 $\vec{y} \mathrel{*}= \vec{y}/z$ |

Table 4: We provide diagrams and code for various forms of SoftMax. $\beta$ is the inverse tempurature parameter, and is set to $d^{-0.5}$.

Fusing SoftMax with a contraction limits the output size. The join tail allows it to be fused with a contraction. This limits the size of the output, yielding a streamable algorithm. To streamline the derivation, we do not explicitly draw the updated maintained variables as in Figure 44. We can then apply rearrangements to recover a streamable form of the composed function.

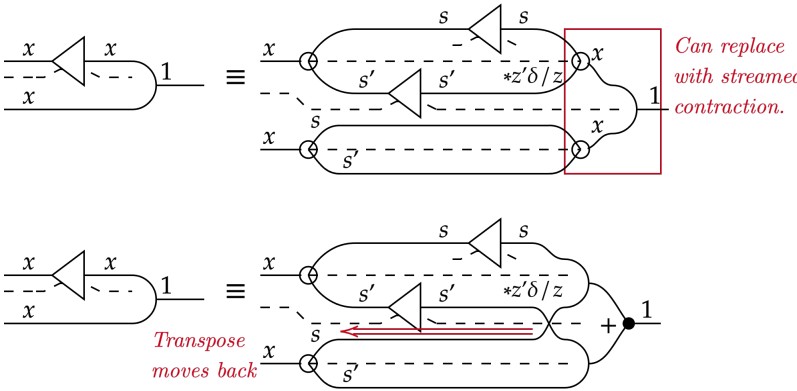

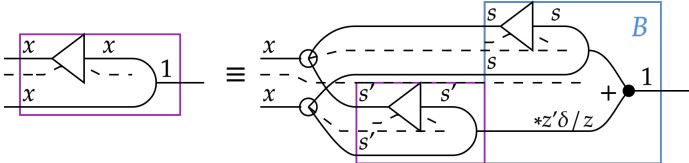

We can then replace SoftMax with unscaled SoftMax, which is done in FlashAttention-2. This lets us move the shared "$/z$" factor to the end of the expression, producing the numerically stable form we use. We forego drawing the auxiliary variables lines to streamline the derivation but add them later.

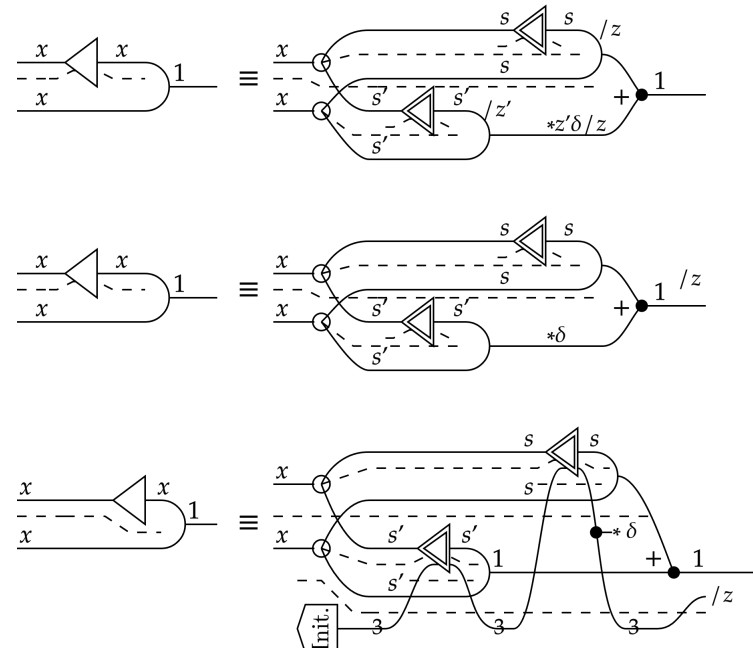

This analysis derives the streamability of SoftMax and its fusion with contraction as a standard procedure using diagrams. Diagrams, by showing the structure of constituent algorithms, act as algebraic tools for deriving fusion.

# B    Performance Analysis

*The performance analysis is assisted by an Excel spreadsheet we developed, available at github.com/mit-zardini-lab/Napkin. In the future, additional tools will be provided at that repository.*

### B.0.1    Lower Bound on Compute Time

The lower bound on compute time is given by the clock cycles per thread for tensor cores $k_{\mathrm{TC}}$ times the number of threads $g_{\overline{q}}$ divided by the clock frequency $f_K$. This must be larger than the transfer time, which is equal to the bytes per iteration $H$ times the per SM bandwidth $B/N_{\mathrm{SM}}$. We assume no caching and a sufficiently large $\overline{x}$. These assessments are given in *Table 4* for different algorithms in the linked document. We artificially set "*Num. of Th*" to 999 for the FlashAttention algorithms to simulate not being memory bound by using caches.

$$T_K \geqslant T_H$$
$$k_{\text{TC}} \ g_{\overline{q}}/f_K \geqslant H \ B/N_{\text{SM}}$$
$$g_{\overline{q}} \geqslant f_K \ H \ B/(f_K \ N_{SM})$$
$$\geqslant 295 \ (\text{for H100 SXM5})$$

### B.1 Arithmetic Intensity of Matrix Multiplication

With FLOPs $K = 2abc$, we are required to perform $M^{-0.5} + (2b)^{-1}$ transfers per FLOP, $H/K$. These computations and transfers occur across the GPU, meaning we use GPU-wide bandwidth. Each compute requires a transfer from GMEM to the L2 cache (at 3352 GB/s) and a transfer from the L2 cache to SMEM (at 12 TB/s) (Shah et al., 2024). This translates to a bandwidth of $1.7e12$ and $6.0e12$ FP16 values per second, respectively. As the L2 cache uses an intermediate caching strategy, its effective memory is given by the total memory of child processors where data is ultimately stored (see Appendix A.2.1). Therefore, we use $M_{L2} = N_{SM} M_{SM}$.

To avoid a bandwidth bottleneck, we require that $M^{-0.5} + (2b)^{-1} < \dot{H}/\dot{K}$. We assemble these values and subsequent calculations in the attached document's "MatMul Bandwidth" sheet. The *absolute minimum* values indicate the $M$ and $b$ values that, if not exceeded, will ensure the algorithm is bottlenecked, independent of other factors. The *Min b* values indicate the size of the inner-dimension we must exceed, given the GPU memory, to not be bandwidth bottlenecked. Note that these expressions give minimums, larger $s_b$ and imperfect caching will slow down the algorithm.

| Bandwidth Limits | | | | |
|---|---|---|---|---|
| FLOPs | | Kt | 9.894E+14 | ops/s |
| Quantization (bytes/value) | | q | 2 | B/value |
| **L2 Level** | This corresponds to the caching level | | | |
| Bandwidth (bytes) | | | 3.35E+12 | B/s |
| Bandwidth (values) | | Ht_L2 | 1.68E+12 | values/s |
| Computes/Transfer | | Kt/Ht | 591 | |
| Absolute Min L2 Memory | | q(Kt/Ht)^2 | 681 | KB |
| Absolute Min b | | (Kt/Ht)/2 | 295 | values |
| N_SM per L2 | | N_SM | 66 | |
| Memory/Processor | | M_SM N_SM | 16896 | KB |
| Tile Size | | (M/q)^0.5 | 2941 | values |
| Min b | (Ht/Kt-(M/q)^-0.5)^-1/2 | | 370 | values |
| **SMEM Level** | This corresponds to the lower level | | | |
| Bandwidth (bytes) | | | 1.20E+13 | B/s |
| Bandwidth (values) | | Ht_SM | 6.00E+12 | values/s |
| Computes/Transfer | | Kt/Ht | 165 | |
| Absolute Min L2 Memory | | q(Kt/Ht)^2 | 53 | KB |
| Absolute Min b | | (Kt/Ht)/2 | 82 | values |
| Memory/Processor | | M_SM | 256 | KB |
| Tile Size | | (M/q)^0.5 | 362 | values |
| Min b | (Ht/Kt-(M/q)^-0.5)^-1/2 | | 151 | values |

## B.2 Configuration Tables

### B.2.1 Derived Attention

**Table 1: Configuration Table for Hopper Attention**

| | Variables | Level | g | w | t | s | u | d | d' | d'' | Multiple | Extra Loaded | Replacement | Quantization (B) | Per Thread Block | Per Warpgroup | Storage |
|---|---|---|---|---|---|---|---|---|---|---|---|---|---|---|---|---|---|
| | | | Partitions (per g_q) | | | | | | | | | | | | | | |
| | *Divisors* | | 128 | 128 | 128 | 32 | 8 | 128 | 16 | 1 | | | | | | | |
| | *Axis Sizes* | | 128 | 128 | 128 | 64 | 64 | 128 | 32 | 8 | | | | | | | |
| Q | Queries | Tensor Core | | 1 | | | | 1 | | | | | | 1 | | 16384 | SMEM |
| K | Keys | Tensor Core | | | | 1 | | 1 | | | | | 1 | 1 | 16384 | | SMEM |
| V | Values | Tensor Core | | | | 1 | | 1 | | | | | 1 | 2 | 32768 | | SMEM |
| S | Attention Scores | Tensor Core | | 1 | | 1 | | | | | | | | 2 | | 16384 | Registers |
| P | Weighted Scores | SMEM | 1 | | | 1 | | | | | | | | 2 | | 16384 | SMEM |
| P' | Weighted Scores (Registers) | Registers | | | 1 | | 1 | | | | | | | 2 | | 16384 | Registers |
| A | Weighted Scores (Tensor Core) | Tensor Core | | 1 | | | | | 1 | | | | | 2 | | 8192 | SMEM |
| α | Auxiliary | Registers | | | 1 | | | | | | 3 | | | 2 | | 768 | Registers |
| O' | Output (Registers) | Registers | | | 1 | | 1 | | | | | | | 2 | | 32768 | Registers |
| D | Delta Output (Tensor Core) | Tensor Core | | 1 | | | | | 1 | | | | | 2 | | 8192 | Registers |
| ΔO | Delta Output (SMEM) | SMEM | 1 | | | | | | 1 | | | | | 2 | | 8192 | SMEM |
| ΔO' | Delta Output (Registers) | Registers | | | 1 | | | | | 1 | | | | 2 | | 2048 | Registers |
| | *Total (Bytes)* | | | | | | | | | | | | | | 49152 | 49152 | SMEM |
| | | | | | | | | | | | | | | | 0 | 76544 | Registers |

*Number of Warpgroups Configuration*

N = 3     g_q = 384

**Table 2: Maximum Analysis**

| | Maximum | Per TB. | Per WG. | N_max | Per TB. | Per WG. | Per Th. |
|---|---|---|---|---|---|---|---|
| | | | | | Excess Calculations | | |
| SMEM | 227 | 48 | 48 | 3.7 | 35 | 11.67 | |
| Registers | 256 | 0 | 74.75 | 3.4 | 31.75 | 10.58 | 85 |
| *Sources* | Under CUDA-Enabled Datacenter Products, Hopper has CC 9.0 | | | | | | |
| | Technical Specifications per Compute Capability | | | | | | |

**Table 3: Compute Analysis**

| | Q-K | SoftMax | P-V | MADD | | Tensor Core | Special Function | FP16 |
|---|---|---|---|---|---|---|---|---|
| Core Type | Tensor Core | Special Function | Tensor Core | FP16 | Totals | | | |
| Ops/Th | 2d s_x | s_x + s_x/u | 2d s_x | 2d s_x / u_x | | | | |
| *From table* | 16384 | 65 | 16384 | 256 | | 32768 | 65 | 256 |
| Ops/Clk | 8192 | 16 | 4096 | 512 | | 5461 | 16 | 512 |
| Clks/Th | 2 | 4.0625 | 4 | 0.5 | | 6 | 4.06 | 0.5 |
| *Sources* | Table , p39. (Ops/clk) = (FLOPs)/(N_SM*f_K) | | | | | | | |
| | CUDA Programming Guide, Arithmetic Instructions | | | | | | | |

**Table 4: Throughput Analysis**

| | | H100 SXM5 | H100 PCIe | Unit |
|---|---|---|---|---|
| Clk/Th | k | 6 | | |
| H/TB | H | 24576 | | |
| Clk/s | f_K | 1.83E+09 | 1.62E+09 | Hz |
| Bandwidth | B | 3.35E+12 | 2.04E+12 | B/s |
| Num. of SMs | N_SM | 132 | 114 | |
| Bandwidth / SM | B / N_SM | 2.54E+10 | 1.79E+10 | B/s |
| Load Time | $T_H = H\, N_{SM} / B$ | 9.68E-07 | 1.37E-06 | s |
| Min. Th | $g^*_q = T_H\, f_K / k$ | 295 | 371 | |
| Num. of Th | g_q | 384 | | |
| Comp Time | $T_K = k * g_q / f_K$ | 1.26E-06 | 1.42E-06 | s |
| Time | $T = \max(T_K, T_H)$ | 1.26E-06 | 1.42E-06 | |
| Ops/loop | $K = g_q * (K / g_q)$ | 1.26E+07 | 1.26E+07 | FLOP |
| Throughput | $K * N_{SM} / T$ | 1.32E+15 | 1.01E+15 | FLOP/s |
| *Sources* | Table, p39 | | | |

### B.2.2  FlashAttention-3 FP8 Inter-Warpgroup

**Table 1: Configuration Table for Hopper Attention**

| Variables | | Level | Partitions (per g_q) | | | | | Multiple | Extra Loaded | Replacement | Quantization (B) | Per Thread Block | Per Warpgroup | Storage |
|---|---|---|---|---|---|---|---|---|---|---|---|---|---|---|
| | | | g | w | t | s | d | | | | | | | |
| | Divisors | | 128 | 1 | 1 | 32 | 128 | | | | | | | |
| | Axis Sizes | | 128 | 1 | | 224 | 128 | | | | | | | |
| Q | Queries | Tensor Core | 1 | | | | 1 | | | | 1 | | 16384 | Registers |
| K | Keys | Tensor Core | | | | 1 | 1 | | 1 | | 1 | 57344 | | SMEM |
| V | Values | Tensor Core | | | | 1 | 1 | | 1 | | 1 | 57344 | | SMEM |
| S | Scores / Weights | Tensor Core | 1 | | | 1 | | | | | 2 | | 57344 | Registers |
| O | Output | Tensor Core | 1 | | | | 1 | | | | 1 | | 16384 | Registers |
| α | Auxiliary | Registers | 1 | | | | | 3 | | | 2 | | 768 | Registers |
| | Total (Bytes) | | | | | | | | | | | 114688 | 0 | SMEM |
| | | | | | | | | | | | | 0 | 90880 | Registers |
| | | | | | | | | | | Excess Registers per Thread | | | 314 | |

**Number of Warpgroups Configuration**

N = 2    g_q = 256

**Table 2: Maximum Analysis**

| | Maximum | Per TB. | Per WG. | N_max | Excess Calculations | | | |
|---|---|---|---|---|---|---|---|---|
| | | | | | Per TB. | Per WG. | Per Th. | |
| SMEM | 227 | 112 | 0 | #DIV/0! | 115 | 57.50 | | |
| Registers | 256 | 0 | 88.75 | 2.9 | 78.5 | 39.25 | 314 | |
| Sources | Under CUDA-Enabled Datacenter Products, Hopper has CC9.0 | | | | | | | |
| | Technical Specifications per Compute Capability | | | | | | | |

**Table 3: Compute Analysis**

| | Q-K | SoftMax | P-V | | | Totals | | |
|---|---|---|---|---|---|---|---|---|
| Core Type | Tensor Core | Special Function | Tensor Core | | Tensor Core | Special Function | | |
| Ops/Th | 2 d s_x | s_x | 2 d s_x | | | | | |
| From table | 57344 | 224 | 57344 | | 114688 | 224 | | |
| Ops/Clk | 8192 | 16 | 8192 | | 8192 | 16 | | |
| Clks/Th | 7 | 14 | 7 | | 14 | 14.00 | | |
| Sources | Table , p39. (Ops/clk) = (FLOPs)/(N_SM*f_K) | | | | | | | |
| | CUDA Programming Guide, Arithmetic Instructions | | | | | | | |

**Table 4: Throughput Analysis**

| | | | H100 SXM5 | H100 PCIe | Unit |
|---|---|---|---|---|---|
| Clk/Th | | k | 14 | | |
| H/TB | | H | 57344 | | B |
| Clk/s | | f_K | 1.83E+09 | 1.62E+09 | Hz |
| Bandwidth | | B | 3.35E+12 | 2.04E+12 | B/s |
| Num. of SMs | | N_SM | 132 | 114 | |
| Bandwidth / SM | | B / N_SM | 2.54E+10 | 1.79E+10 | B/s |
| Load Time | T_H = H N_SM / B | | 2.26E-06 | 3.21E-06 | s |
| Min. Th | g*_q = T_H f_K / k | | 295 | 371 | |
| Num. of Th | | g_q | 999 | | |
| Comp Time | T_K = k * g_q / f_K | | 7.64E-06 | 8.63E-06 | s |
| Time | T = max(T_K, T_H) | | 7.64E-06 | 8.63E-06 | |
| Ops/loop | K = g_q * (K / g_q) | | 1.15E+08 | 1.15E+08 | FLOP |
| Throughput | K * N_SM / T | | 1.98E+15 | 1.51E+15 | FLOP/s |
| Sources | Table , p39 | | | | |

### B.2.3  FlashAttention-3 FP8 Intra-Warpgroup

**Table 1: Configuration Table for Hopper Attention**

| Variables | | Level | Partitions (per g_q) | | | | Multiple | Extra Loaded | Replacement | Quantization (B) | Per Thread Block | Per Warpgroup | Storage |
|---|---|---|---|---|---|---|---|---|---|---|---|---|---|
| | | | g | | s | d | | | | | | | |
| | Divisors | | 128 | 1 | 1 | 32 | 128 | | | | | | |
| | Axis Sizes | | 128 | | 192 | 128 | | | | | | | |
| Q | Queries | Tensor Core | 1 | | | 1 | | | | 1 | | 16384 | Registers |
| K | Keys | Tensor Core | | | 1 | 1 | | 1 | | 1 | 49152 | | SMEM |
| V | Values | Tensor Core | | | 1 | 1 | | 1 | | 1 | 49152 | | SMEM |
| S | Scores / Weights | Tensor Core | 1 | | 1 | | | 1 | | 2 | | 98304 | Registers |
| O | Output | Tensor Core | 1 | | | 1 | | | | 1 | | 16384 | Registers |
| α | Auxiliary | Registers | 1 | | | | 3 | | | 2 | | 768 | Registers |
| | Total (Bytes) | | | | | | | | | | 98304 | 0 | SMEM |
| | | | | | | | | | | | 0 | 131840 | Registers |
| | | | | | | | | | Excess Registers per Thread | | | -6 | |

**Number of Warpgroups Configuration**

| N = | 1 | | g_q = | 128 | | | |
|---|---|---|---|---|---|---|---|

**Table 2: Maximum Analysis**

| | Maximum | Per TB. | Per WG. | N_max | Per TB. | Per WG. | Per Th. |
|---|---|---|---|---|---|---|---|
| | | | | Excess Calculations | | | |
| SMEM | 227 | 96 | 0 | #DIV/0! | 131 | 131.00 | |
| Registers | 256 | 0 | 128.75 | 2.0 | 127.25 | 127.25 | 1018 |
| **Sources** | Under CUDA-Enabled Datacenter Products, Hopper has CC9.0 | | | | | | |
| | Technical Specifications per Compute Capability | | | | | | |

**Table 3: Compute Analysis**

| | | Q-K | SoftMax | P-V | | | Totals | |
|---|---|---|---|---|---|---|---|---|
| Core Type | | Tensor Core | Special Function | Tensor Core | | Tensor Core | Special Function | |
| Ops/Th | | 2 d s_x | s_x | 2 d s_x | | | | |
| From table | | 49152 | 192 | 49152 | | 98304 | 192 | |
| Ops/Clk | | 4096 | 16 | 4096 | | 4096 | 16 | |
| Clks/Th | | 12 | 12 | 12 | | 24 | 12.00 | |
| **Sources** | Table , p39. (Ops/clk) = (FLOPs)/(N_SM*f_K) | | | | | | | |
| | CUDA Programming Guide, Arithmetic Instructions | | | | | | | |

**Table 4: Throughput Analysis**

| | | | H100 SXM5 | H100 PCIe | Unit |
|---|---|---|---|---|---|
| Clk/Th | | k | 24 | | |
| H/TB | | H | 49152 | | B |
| Clk/s | | f_K | 1.83E+09 | 1.62E+09 | Hz |
| Bandwidth | | B | 3.35E+12 | 2.04E+12 | B/s |
| Num. of SMs | | N_SM | 132 | 114 | |
| Bandwidth / SM | | B / N_SM | 2.54E+10 | 1.79E+10 | B/s |
| Load Time | $T\_H = H N\_SM / B$ | | 1.94E-06 | 2.75E-06 | s |
| Min. Th | $g^*\_q = T\_H f\_K / k$ | | 148 | 185 | |
| Num. of Th | | g_q | 999 | | |
| Comp Time | $T\_K = k * g\_q / f\_K$ | | 1.31E-05 | 1.48E-05 | s |
| Time | $T = max(T\_K, T\_H)$ | | 1.31E-05 | 1.48E-05 | |
| Ops/loop | $K = g\_q * (K / g\_q)$ | | 9.82E+07 | 9.82E+07 | FLOP |
| Throughput | $K * N\_SM / T$ | | 9.89E+14 | 7.56E+14 | FLOP/s |
| **Sources** | Table, p39 | | | | |

## B.2.4 FlashAttention-3 FP16 Intra-Warpgroup

**Table 1: Configuration Table for Hopper Attention**

| | Variables | Level | g | | | s | d | Multiple | Extra Loaded | Replacement | Quantization (B) | Per Thread Block | Per Warpgroup | Storage |
|---|---|---|---|---|---|---|---|---|---|---|---|---|---|---|
| | | | Partitions (per g_q) | | | | | | | | | | | |
| | **Divisors** | | 128 | 1 | 1 | 32 | 128 | | | | | | | |
| | **Axis Sizes** | | 128 | | | 128 | 128 | | | | | | | |
| Q | Queries | Tensor Core | 1 | | | | 1 | | | | 2 | | 32768 | Registers |
| K | Keys | Tensor Core | | | | 1 | 1 | 1 | | 1 | 2 | 65536 | | SMEM |
| V | Values | Tensor Core | | | | 1 | 1 | 1 | | 1 | 2 | 65536 | | SMEM |
| S | Scores / Weights | Tensor Core | 1 | | | 1 | | 1 | | 1 | 2 | | 65536 | Registers |
| O | Output | Tensor Core | 1 | | | | 1 | | | | 2 | | 32768 | Registers |
| α | Auxiliary | Registers | 1 | | | | | 3 | | | 4 | | 1536 | Registers |
| | Total (Bytes) | | | | | | | | | | | 131072 | 0 | SMEM |
| | | | | | | | | | | | | 0 | 132608 | Registers |
| | | | | | | | | | | | Excess Registers per Thread | | -12 | |

**Number of Warpgroups Configuration**

| N = | 1 | | g_q = | 128 | | | |
|---|---|---|---|---|---|---|---|

**Table 2: Maximum Analysis**

| | Maximum | Per TB. | Per WG. | N_max | Per TB. | Per WG. | Per Th. |
|---|---|---|---|---|---|---|---|
| | | | | Excess Calculations | | | |
| SMEM | 227 | 128 | 0 | #DIV/0! | 99 | 99.00 | |
| Registers | 256 | 0 | 129.5 | 2.0 | 126.5 | 126.50 | 1012 |
| **Sources** | Under CUDA-Enabled Datacenter Products, Hopper has CC9.0 | | | | | | |
| | Technical Specifications per Compute Capability | | | | | | |

**Table 3: Compute Analysis**

| | | Q-K | SoftMax | P-V | | | Totals | |
|---|---|---|---|---|---|---|---|---|
| Core Type | | Tensor Core | Special Function | Tensor Core | | Tensor Core | Special Function | |
| Ops/Th | | 2 d s_x | s_x | 2 d s_x | | | | |
| From table | | 32768 | 128 | 32768 | | 65536 | 128 | |
| Ops/Clk | | 4096 | 16 | 4096 | | 4096 | 16 | |
| Clks/Th | | 8 | 8 | 8 | | 16 | 8.00 | |
| **Sources** | Table , p39. (Ops/clk) = (FLOPs)/(N_SM*f_K) | | | | | | | |
| | CUDA Programming Guide, Arithmetic Instructions | | | | | | | |

**Table 4: Throughput Analysis**

| | | | H100 SXM5 | H100 PCIe | Unit |
|---|---|---|---|---|---|
| Clk/Th | | k | 16 | | |
| H/TB | | H | 65536 | | B |
| Clk/s | | f_K | 1.83E+09 | 1.62E+09 | Hz |
| Bandwidth | | B | 3.35E+12 | 2.04E+12 | B/s |
| Num. of SMs | | N_SM | 132 | 114 | |
| Bandwidth / SM | | B / N_SM | 2.54E+10 | 1.79E+10 | B/s |
| Load Time | $T\_H = H N\_SM / B$ | | 2.58E-06 | 3.66E-06 | s |
| Min. Th | $g^*\_q = T\_H f\_K / k$ | | 295 | 371 | |
| Num. of Th | | g_q | 999 | | |
| Comp Time | $T\_K = k * g\_q / f\_K$ | | 8.73E-06 | 9.87E-06 | s |
| Time | $T = max(T\_K, T\_H)$ | | 8.73E-06 | 9.87E-06 | |
| Ops/loop | $K = g\_q * (K / g\_q)$ | | 6.55E+07 | 6.55E+07 | FLOP |
| Throughput | $K * N\_SM / T$ | | 9.89E+14 | 7.56E+14 | FLOP/s |
| **Sources** | Table, p39 | | | | |

