# OpenReview forum: "FlashAttention on a Napkin: A Diagrammatic Approach to Deep Learning IO-Awareness"
_TMLR — Accepted by TMLR_

### Review · Reviewer_rQeH · 2024-12-19

**Summary Of Contributions:**

This paper introduces a diagrammatic representation of Deep Learning (DL) algorithms. It defines rules and building blocks of this approach, and connects them to implement operations and algorithms of various complexity, offering estimates of memory usage (across memory levels) and transfer costs. Such algorithms are then expanded into pseudo-code with explicit representation of loops and variables, allowing for a direct comparison with existing algorithms like FlashAttention. Using the proposed approach, the authors derive two attention algorithms, for Ampere and Hopper architectures, which may theoretically outperform conventional attention algorithms.

**Audience:**

Yes

**Broader Impact Concerns:**

No concern on my end.

**Claims And Evidence:**

Yes

**Requested Changes:**

I have a positive view of this paper in its present form but I would appreciate if the authors could address my comments in the other section.

**Strengths And Weaknesses:**

**strengths**
- the paper is very well written, the approach is introduced step by step, each section building on the definitions and insights of the previous one, towards progressively more complex algorithms
- the topic of diagrammatic representation of DL algorithms can be of interest to at least part of the audience of this journal, and prompt further work in this area
- to the best of my knowledge, the proposed approach is novel and unique
- the authors show that thanks to their framework they are able to extract useful insights from existing algorithms and craft hardware-specific alternatives with the potential to outperform existing strategies

**weaknesses**
- the main drawback is the lack of on-hardware verification of the theoretical estimates for memory consumption and transfer costs derived using the proposed framework, as well as a lack of demonstration of the efficacy of the newly derived attention algorithms

**other comments**
- in Figure 12: what is the factor $N_g$ appearing in the cumulative per-group transfer cost $H_{l1,g}$ formula?
- Figure 15 is not referenced in the text
- in Section 3.1 and Figure 18, could you spell out the derivation of the memory consumption inequalities for matmul ($\sqrt M \ge g_c$) and why "the transfer cost of (...) matrix multiplication is cubic for $n \ge \sqrt M/2$". Also, why is it assumed that a = b = c, isn't this a loss of generality on the shape of matrices?
- Section 4.3 deals with values expressed with fewer bits, and comes to the natural conclusion that quantization is beneficial for throughput (everything else being equal). However, quantization algorithms typically rely on storing additional quantization parameters (such as scaling factor) that are used to quantize / dequantize the matrix. This represent an overhead in terms of memory consumption, transfer costs, as well as operations. For example, for a matrix of size N x M and group size GS, group-quantization stores (N / GS) x M scales, twice as many if asymmetric quantization is enabled. It would be interesting expanding this section to consider the whole impact of some actual quantization algorithms, like the aforementioned per-group. As presented, the finding of this section appears to be trivial

---

### Review · Reviewer_JqHN · 2025-01-01

**Summary Of Contributions:**

The paper provides a diagrammatic framework, based on Neural Circuit Diagrams for representing and analyzing algorithms focusing on hardware constraints and optimization. Through this method it proposes a theoretical optimization to FlashAttention that suggests potential for improved hardware utilization, though it remains untested.

**Audience:**

Yes

**Broader Impact Concerns:**

No concerns.

**Claims And Evidence:**

Yes

**Requested Changes:**

1. I would like to see more clarity about what this framework can and cannot do. If it's primarily an analytical tool rather than a method for discovering new algorithms, the paper should be more precise in its claims.

2. The paper would benefit from comparing it with existing performance modeling tools (if any exist). Why should researchers invest time in learning such method? Some concrete comparisons would help make the case.

3. The paper is quite dense and challenging to follow. More intuitive examples and clearer explanations of hardware concepts would help readers without extensive hardware backgrounds.

4. Most importantly, I think some empirical validation would greatly strengthen the paper. Is it feasible to implement and test some of their proposed optimizations? Concrete benchmarks would make the theoretical contributions more convincing. This paper is already difficult to follow with a steep learning curve, but demonstrating its practicality and effectiveness could convince engineers and researchers to adopt this method.

**Strengths And Weaknesses:**

This is a highly unusual technical paper with loads of details, and I can only imagine the immense amount of work put into such paper. The idea of using diagrams to visualize and analyze algorithms is intriguing and could potentially lead to better understanding of problems and further enhancing the solutions.

Weaknesses (Not all of the following are necessarily weaknesses, but rather things that are not entirely clear to me and areas of confusion that could be addressed or clarified):

First, I find myself struggling with the paper's claim that it can be "used to quickly derive methods like FlashAttention." To my best understsanding of the paper, the approach seems primarily post-hoc - going from known algorithm -> diagram representation -> analysis, rather than going from problem -> diagram analysis -> discover new algorithm. While the analysis is thorough, I'm not convinced that someone unfamiliar with FlashAttention could discover similar optimizations using just this diagrammatic method.

This raises a more fundamental question: what is the value proposition if we already understand algorithms like FlashAttention? The method appears capable of suggesting theoretical optimizations - for instance, fitting 13 warps per SM instead of FlashAttention's 4-8. It seems that it may be capable of suggesting theoretical optimizations to further optimize approaches that are already known. But without empirical validation, it's hard to judge whether these optimizations would work in practice. I think the paper lacks empirical rigor despite being highly technical and information dense.

The learning curve also concerns me. The framework requires understanding both the diagrammatic notation and hardware concepts. I wonder if the complexity is justified by the analytical insights gained. While I can see its utility as an analytical tool for IO-aware optimization, I'm less convinced about its broader adoption given the investment required to use it effectively.

---

### Review · Reviewer_UB1n · 2025-01-02

**Summary Of Contributions:**

This paper presents a diagrammatic framework for optimizing deep learning algorithms, focusing on IO-awareness to address memory bandwidth and transfer bottlenecks in GPU-based computations. It proposes a performance model based on diagrams that depict data types, functions, and GPU hierarchies, allowing for systematic derivation of optimized algorithms. It also introduces techniques such as group partitioning and stream partitioning for resource-efficient computations. Additionally, the authors delve into hardware-specific optimizations, such as coalesced memory access and tensor-core operations, presenting pseudocode tailored for Ampere and Hopper architectures.

**Audience:**

Yes

**Claims And Evidence:**

Yes

**Requested Changes:**

- Please include experimental benchmarks comparing the proposed scheme with SOTA methods like FlashAttention-3. At a minimum, provide empirical validation under basic settings to demonstrate performance gains. Please report metrics such as execution time, and memory usage.
- The process of understanding the calculations demonstrating that the proposed framework can fit up to 13 warps per thread block, as shown in Table 3, is overly complex and scattered across the appendix. It would be more effective to present the key computational steps clearly on the main pages for easier comprehension.
- Please ensure consistent notation such as $$(\mathbb{R}^{a \times b \times c} \times \mathbb{R}^{d \times e \times c}) \rightarrow (\mathbb{R}^{a \times b \times c} \times \mathbb{R}^{d \times e}) $$ in Section 2.1.

**Strengths And Weaknesses:**

### Strengths
- The diagrammatic approach tries to simplify complex deep learning algorithm optimizations, making them more accessible and systematic.
- The paper combines theoretical insights with practical hardware considerations, offering a holistic view of algorithm optimization for GPUs.

### Weaknesses
- The framework lacks experimental results to validate its theoretical performance improvements. At the very least, empirical validation under minimal settings should be provided for methods like FlashAttention-3 to substantiate the claims.
- While diagrams are central to the contribution, some are dense and may overwhelm readers unfamiliar with the methodology. More intuitive explanations or detailed walkthroughs are needed.
- The lack of guidelines for creating diagrams may lead to inconsistent representations across researchers, limiting the framework's utility.

---

### Author Response · Authors · 2025-01-05
**Response to Reviewer Comments**

Thank you for your reviews. We were surprised by how quickly reviews came in and are glad to see the support. We will make modifications and upload the updated paper soon.

We will restructure the paper to emphasise the methodology we present rather than the possible success of the proposed algorithms. The paper's main point is to present a framework that can systematically and quickly derive algorithms and associated performance models. This addresses the major challenge of rapidly developing algorithms for the latest hardware.

The algorithms in Section 5 aim to show that a logical procedure can derive hardware-aware algorithms, not to present incrementally improved attention algorithms with associated empirical results. The algorithms are derived by a step-by-step procedure and differ from FlashAttention in key ways that show diagrams are not merely a post-hoc rationalization for existing methods. We will deemphasise the claims that these algorithms deliver superior performance and better explain this systematic procedure, moving content away from the appendix.

Papers with [interesting findings rather than achieving state-of-the-art performance](https://jmlr.org/tmlr/acceptance-criteria.html) are the emphasis of TMLR. Even if the proposed algorithms are somewhat slower than state-of-the-art methods, their ability to be derived in days compared to the years that FlashAttention required is their major point of interest. This is especially the case given the upcoming release of Blackwell (likely within the week), which will mean Hopper is no longer state-of-the-art and will generate interest in methods that can quickly develop kernels.

**Major Changes.**
- **Emphasise the main result of the paper.** Diagrams can be used to quickly derive optimizations for algorithms, and generate an extensive performance model. This, more than some incremental improvement, is the most intriguing aspect of the work.
- **Elaborate on the methodology of Section 5.** Show how given a function we aim to implement and hardware capabilities, the algorithms we present logically follow. More carefully lay out the step-by-step process, moving content away from the appendix. Emphasise that this is the paper's main contribution, and elaborate how this methodology can be adapted for other architectures.

**Minor Changes.**
- Make the notation more consistent.
- Explain hardware concepts in greater detail.
- Elaborate on quantization scaling and matrix multiplication.

---

### Author Response · Authors · 2025-01-15
**Major Changes: Focusing on Methodology, Provide Analysis of FlashAttention**

The previous version of the paper attempted to conform to the page limit and have its primary contribution be a potentially improved attention algorithm. With the added length of a long submission, we have focused on fleshing out the theoretical framework, the importance of which is better supported by the content of the paper.

**Major Changes**
- The claims of the paper have been adjusted from "*a potentially improved algorithm*" to "*providing a framework for relating assumptions about GPU behavior to claims about performance*". This contribution is better supported by the collection of representational schemes, performance models, derivation procedures, and analytical tools present in the diagram.
- Section 5 has been fleshed out to focus on the step-by-step derivation of an algorithm for a toy Hopper hierarchy. This methodology allows unincorporated Hopper features (manipulating fragmented memory) to be added in the future and adaptations to be made for other architectures. We better explain how subloops are identified, how configuration tables are created, and how clock-cycle analysis is performed.
- Section 6 has been added, where we apply the tools from Section 5 to analyze FlashAttention. We comment on how the assumptions of our framework are contradicted and supported by its performance, guiding future improvement.
- We have added a document that includes configuration tables for various algorithms, including our own and three forms of FlashAttention-3.

**Minor Changes**
- We have expanded the analysis of matrix multiplication to show how diagrams relate to tiling strategies.
- On page 8, matrix multiplication and the relationship between diagrams and tiling strategies is covered in greater depth.

**Current State of the Paper**

In its current state, the paper contributes:
- A diagrammatic scheme for representing algorithms and the application of optimization strategies supported by theorems regarding the compositional properties of fused algorithms. This improves on prior methods by making the high-level derivation of strategies like FlashAttention almost trivial, simply requiring a relabeling.
- An in-depth performance model that considers multi-level hierarchies and the variable sensitivity $\beta$ of algorithms to lower-level memory sizes. This improves on naive performance models that neglect lower-level memory sizes, which are typically used for matrix multiplication. In the case of FlashAttention, its performance model refers to a generic $M$, which we show to correspond to the lowest memory size where output data is stored or to the collective lowest level memory size $M N$ for an intermediate caching level.
- A method for expanding high-level diagrams into intermediate-level pseudocode, accommodating hardware features like coalesced memory access and tensor core operations.
- A method for calculating required clock cycles from diagrams, allowing overlapping strategies and the impact of subalgorithm overhead to be considered.
- An analysis of FlashAttention and the observation that the algorithm is restricted by the slow special function unit required for the SoftMax operation.

Overall, these methods contribute a systematic framework for relating hypotheses about GPU behaviour to claims about performance. In many cases, these claims are absolute. A GPU can only perform so many tensor cores or special function units per clock cycle, and algorithms require a certain amount of memory to be stored and have an absolute minimum number of required transfers. Differences between the model and realized performance comes down to *overhead*, which can be incorporated by future empirical testing.

We believe that empirical work is best left for the future. The paper already has substantial scope and contributions of interest to readers. It addresses several gaps in the research: the difficulty of discovering optimization strategies, and the weakness of prevailing performance models. Furthermore, separating theoretical and empirical work provides confidence that claims are not post-hoc rationalizations of a few successful experiments, strengthening scientific value. As new architectures are released, the long-term value of this paper will be in the framework it provides which allows for iterative improvement of our understanding, rather than the specific claims it makes about Hopper.

---

### Decision · Action_Editor_ogaq · 2025-02-12

**Recommendation:** Accept with minor revision

**Comment:**

Overall this is an interesting and comprehensive work introducing a novel framework for analyzing algorithms running on a GPU and their performance characteristics. All reviewers appreciated this novelty and the technical contribution of this paper, as well as the use of FlashAttention to illustrate the framework's potential utility. While it would have been nice to provide empirical validation to really demonstrate the utility of the framework, as it is, the contribution of the diagrammatic analysis framework already satisfies TMLR's criteria for acceptance.

Please revise the paper accordingly to update the references (e.g. (?) on Page 18, including publication years, and venues instead of arXiv for published works).

**Audience:**

This paper would be of interest to many in the TMLR audience who are interested in analyzing and improving the performance of deep learning (and other) algorithms implemented on GPUs.

**Claims And Evidence:**

The paper proposes a conceptual,  diagrammatic framework for representing algorithms that run on GPUs, which also considers their low-level hardware architecture. It aims to provide a systematic framework for understanding the performance of these algorithms and to derive optimized algorithms accordingly.

The framework is discussed extensively in the revised paper, with detailed examples showcasing how it can be used to provide insight into commonly used optimized algorithms such as FlashAttention to reveal further areas for improvement. The performance models supported by the framework are richer than existing ones. All reviewers appreciated the contributions of the framework to analyzing and potentially improving algorithms, though they also highlighted the lack of empirical validation of these findings; the authors reduced their claims of providing potentially improved algorithms in the revised paper in response.

The revised claim that the paper provides a framework for relating algorithms' performance in the context of GPU modelling assumptions is well-supported by the current version of the paper, as it describes clearly such a methodology, illustrating it with examples.